# Insights into Soil NO Emissions and the Contribution to Surface Ozone Formation in China

Ling Huang[1,2], Jiong Fang[1,2], Jiaqiang Liao[1,2], Greg Yarwood[3], Hui Chen[1,2], Yangjun Wang[1,2], Li Li[1,2*]

[1]School of Environmental and Chemical Engineering, Shanghai University, Shanghai, 200444, China

[2]Key Laboratory of Organic Compound Pollution Control Engineering (MOE), Shanghai University, Shanghai, 200444, China

[3]Ramboll, Novato, California, 94945, USA

Correspondence: Li Li (lily@shu.edu.cn)

**Keywords:** Soil NO emissions; Ground-level ozone; BDSNP; OSAT

**Abstract.** Elevated ground-level ozone concentrations have emerged as a major environmental issue in China. Nitrogen oxide ($NO_x$) is a key precursor to ozone formation. Although control strategies aimed at reducing $NO_x$ emissions from conventional combustion sources are widely recognized, soil $NO_x$ emissions (mainly as NO) due to microbial processes have received little attention. The impact of soil NO emissions on ground-level ozone concentration is yet to be evaluated. This study estimated soil NO emissions in China using the Berkeley-Dalhousie soil $NO_x$ parameterization (BDSNP) algorithm. A typical modeling approach was used to quantify the contribution of soil NO emissions to surface ozone concentration. The Brute-force method (BFM) and the Ozone Source Apportionment Technology (OSAT) implemented in the Comprehensive Air Quality Model with extensions (CAMx) were used. The total soil NO emissions in China for 2018 were estimated to be 1157.9 Gg N, with an uncertainty range of 715.7~1902.6 Gg N. Spatially, soil NO emissions are mainly concentrated in Central China, North China, Northeast China, northern Yangtze River Delta (YRD) and eastern Sichuan Basin, with distinct diurnal and monthly variations that are mainly affected by temperature and the timing of fertilizer application. Both the BFM and OSAT results indicate a substantial contribution of soil NO emissions to the maximum daily 8-hour (MDA8) ozone concentrations by 8~12.5 $\mu g/m^3$ on average for June 2018, with the OSAT results consistently higher than BFM. The results also showed that soil NO emissions led to a relative increase in ozone exceedance days by 10%~43.5% for selected regions. Reducing soil NO emissions resulted in a general decrease in monthly MDA8 ozone concentrations, and the magnitude of ozone reduction became more pronounced with increasing reductions. However, even with complete reductions in soil NO emissions, approximately 450.3 million people are still exposed to unhealthy ozone levels, necessitating

multiple control policies at the same time. This study highlights the importance of soil NO
emissions for ground-level ozone concentrations and the potential of reducing NO emissions
as a future control strategy for ozone mitigation in China.
**1. Introduction**
A substantial decrease in the atmospheric fine particulate matter ($PM_{2.5}$) concentrations has
been witnessed during the past decade in China (Zhai et al., 2019; Xiao et al., 2020; Maji,
2020) while the ground-level ozone ($O_3$) concentrations do not exhibit a steady downward
trend (Lu et al., 2020; Lu et al., 2021; Wang et al., 2022a; Sun et al., 2021). Because high
ozone concentration increases respiratory and circulatory risks (Malley et al., 2017; Cakaj et
al., 2023; Wang et al., 2020) and reduces crop yields (Feng et al., 2019; Lin et al., 2018;
Mukherjee et al., 2021; Montes et al., 2022), the coordinate control of $PM_{2.5}$ and $O_3$ was
proposed as part of the $14^{th}$ Five-year plan (Council, 2021). A continuous increase in
summertime surface ozone was observed across China's nationwide monitoring network from
2013 to 2019, followed by an unprecedented decline in 2020 (except for Sichuan Basin) (Sun
et al., 2021), which is equally attributed to meteorology and anthropogenic emissions
reductions (Yin et al., 2021). As a secondary air pollutant, ozone is generated by the
photochemical oxidation of volatile organic compounds (VOC) in the presence of nitrogen
oxides ($NO_x = NO + NO_2$), both of which are considered ozone precursors. The non-linear
response of ozone formation to its precursors is well established (Kleinman et al., 1994;
Sillman et al., 1990). In regions classified as $NO_x$-limited, reducing $NO_x$ emissions is an
effective strategy for ozone mitigation. However, in regions classified as VOC-limited,
typically characterized by high $NO_x$ emissions such as metropolitan areas, decreasing $NO_x$
emissions may actually result in increased ozone concentrations due to reduced ozone titration
by NO and diminished OH titration by $NO_2$ (Seinfeld and Pandis, 2016). Under such
circumstances, reducing VOC emissions will counteract ozone increases caused by reducing
$NO_x$ emissions. The control strategies to mitigate ozone pollution in China focused on
reducing $NO_x$ emissions at an early stage and started to stress the control of VOCs emissions
in recent years (e.g., the 2020 action plan on VOCs mitigation), including control of fugitive
emissions, stringent emissions standards, and substituting raw materials with low VOCs
content (Ecology, 2020). Ding et al. (2021) concluded that for North China Plain (NCP), a
region that experienced the most severe $PM_{2.5}$ and ozone pollution in China, reductions in
$NO_x$ emissions are essential regardless of VOC reduction.
Existing control strategies for $NO_x$ emissions are almost exclusively targeted at combustion
sources, for example, power plants, industrial boilers, cement production, and vehicle
exhausts (Sun et al., 2018; Ding et al., 2017; Diao et al., 2018). However, $NO_x$ emissions from
soils (mainly as NO), as a result of microbial processes (e.g., nitrification and denitrification),

could make up a substantial fraction of the total $NO_x$ emissions (Lu et al., 2021; Drury et al., 2021), yet is often overlooked. In California, soil $NO_x$ emissions in July accounted for 40% of the state's total $NO_x$ emissions (when using an updated estimation algorithm) and resulted in 23% of enhanced surface ozone concentration (Sha et al., 2021). However, a wide range of annual soil $NO_x$ emissions from 8,685 tons (as $NO_2$, (Guo et al., 2020) to 161,100 metric tons of $NO_x$-N (Almaraz et al., 2018) were reported depending on different methods. Romer et al. (2018) estimated that nearly half of the increase in hot-day ozone concentration in a forested area of the rural southeastern United States is attributable to the temperature-induced increases in $NO_x$ emissions, mostly likely due to soil microbes.

Soil NO emissions are affected by many factors, including nitrogen fertilizer application, soil organic carbon content, soil temperature, humidity, and pH (Vinken et al., 2014; Yan et al., 2005; Wang et al., 2021; Skiba et al., 2021). The amount of nitrogen fertilizer application in China was estimated to account for one-third of the global nitrogen fertilizer application (Heffer and Prud'homme, 2016), with most of the land under high nitrogen deposition (Liu et al., 2013; Lü and Tian, 2007). Therefore, soil NO emissions in China are expected to be significant, and their impacts on ozone pollution need to be systematically evaluated. So far, only a limited number of studies have addressed this issue in China (Lu et al., 2021; Shen et al., 2023; Wang et al., 2008; Wang et al., 2022b). Lu et al. (2021) concluded that soil NO significantly reduced the ozone sensitivity to anthropogenic emissions in NCP, therefore, causing a so-called "emissions control penalty". Wang et al. (2022b) reported $NO_x$ emissions from cropland contributed 5.0% of the maximum daily 8h average ozone (MDA8 $O_3$) and 27.7% of $NO_2$ concentration in NCP. These studies focused solely on NCP, a region with persistent $O_3$ pollution in warm seasons (Liu et al., 2020; Lu et al., 2020). The impact of soil NO emissions on ozone concentrations over other regions, for example, the northern Yangtze River Delta (YRD) and Sichuan Basin, where soil emissions are high (see Section 3.1) and ozone pollution is also severe (Shen et al., 2022; Yang et al., 2021), has not been much evaluated in details (Shen et al., 2023). In addition, the method employed in existing studies to evaluate soil NO emissions on ozone concentration is the conventional "brute-force" zero-out approach, which might be inappropriate given the strong nonlinearity of the ozone chemistry (Clappier et al., 2017; Thunis et al., 2019).

With the deepening of emissions control measures for power, industrial and on-road sectors, anthropogenic $NO_x$ emissions from combustion sources have decreased at a much faster rate (by 4.9% since 2012) than that from soil (fertilizer application decreases at a rate of 1.5% since 2015, Fig. S1). Therefore, understanding the impacts of soil NO emissions on ground-level ozone concentration, particularly considering the spatial heterogeneities over different regions of China, is of great importance for formulating future ozone mitigation strategies. In this study, soil NO emissions in China for 2018 were estimated based on a most recent soil

NO parameterization scheme with updated fertilizer data as input. The spatial and temporal variations of soil NO emissions were described first. Uncertainties associated with estimation of soil NO emissions were discussed. An integrated meteorology and air quality model was applied to quantify the impact of soil NO emissions on surface ozone concentration based on two different methods. Lastly, we evaluated the changes in ozone concentration and exposed population under different emission scenarios to highlight the effectiveness of reducing soil NO emissions as potential control policy. Our results provide insights into developing effective emissions reduction strategies to mitigate the ozone pollution in China.

## 2. Methodology

2.1. Estimation of soil NO emissions in China

Soil NO emissions were estimated based on the Berkeley-Dalhousie Soil $NO_x$ Parameterization (BDSNP) that is implemented in the Model of Emissions of Gases and Aerosols from Nature (MEGAN) version 3.2 (https://bai.ess.uci.edu/megan/data-and-code, accessed on September 1[st], 2021). The BDSNP algorithm estimates the soil NO emissions by adjusting a biome-specific NO emissions factor in response to various conditions, including the soil temperature, soil moisture, precipitation-induced pulsing, and a canopy reduction factor (Eq. 1, (Rasool et al., 2016):

$$NO_{\text{emission flux}} = A'_{biome}(N_{avail}) \times f(T) \times g(\theta) \times P(l_{dry}) \times CRF(LAI, Biome, Meterology) \qquad \text{Eq. 1}$$

where $f(T)$ and $g(\theta)$ is the temperature ($T$, unit: K) and soil moisture ($\theta$, unit: $m^3/m^3$) dependence functions, respectively; $P(l_{dry})$ represents the pulsed soil emissions due to wetting of dry soils; $l_{dry}$ (hours) is the antecedent dry period of a pulse; and CRF describes the canopy reduction factor, which is a function of the leaf area index (LAI, $m^2/m^2$) and the meteorology. $A'_{biome}$ (ng N $m^{-2}$ $s^{-1}$) is the biome-specific emission factor, which is further calculated as Eq.2:

$$A'_{biome} = A_{w,biome} + N_{avail} \times \bar{E} \qquad \text{Eq. 2}$$

In Eq. 2, $A_{w,biome}$ (ng N $m^{-2}$ $s^{-1}$) is the wet biome-dependent emission factor; $N_{avail}$ is the available nitrogen from fertilizer and deposition; $\bar{E}$ is the emission rate based on an observed global estimates of fertilizer emissions ((Rasool et al., 2016). The detailed expressions of these parameters are presented in the Supporting Information. More information on the BDSNP parameterizations can be found in previous studies (Hudman et al., 2012).

The default N fertilizer input data provided with the BDSNP algorithm is based on the a (Potter et al., 2010), which gives a number of 19.6 Tg N/a. In this study, we collected fertilizer data from statistical yearbooks at the provincial level. The total amount of pure nitrogen fertilizer (hereafter N fertilizer) applied in the year 2018 is 20.7 Tg N/a, which is similar (5.6% higher) to IFA value. However, besides the N fertilizer, NPK compound fertilizer

(containing nitrogen (N), phosphorous (P), and potassium (K)) is being increasingly applied
in China. According to the statistical yearbook, the amount of N fertilizer applied decreased
from 23.5 Tg in 2010 to 20.7 Tg in 2018 (a relative reduction of 11.9%). In contrast, NPK
fertilizer increased from 18.0 in 2010 to 22.7 Tg in 2018 (a relative increase of 26.1%). We
assumed one-third of the NPK fertilizer is nitrogen (Liu, 2016); thus, the total amount of
nitrogen applied as fertilizer is 28.2 Tg N in 2018, which is 43.9% higher than the value from
Potter et al. (2010). We divided China into seven regions for emission analysis at regional
scale, namely Northeast China, North China, Central China, East China, South China,
Southwest China, and Northwest China, as indicated by different colors in Fig. 1 (see Table
S1 for the list of provinces in each region). At the regional level, the amount of total fertilizer
differs by as much as 9.1% to 46.4% from the default fertilizer (Table S2).

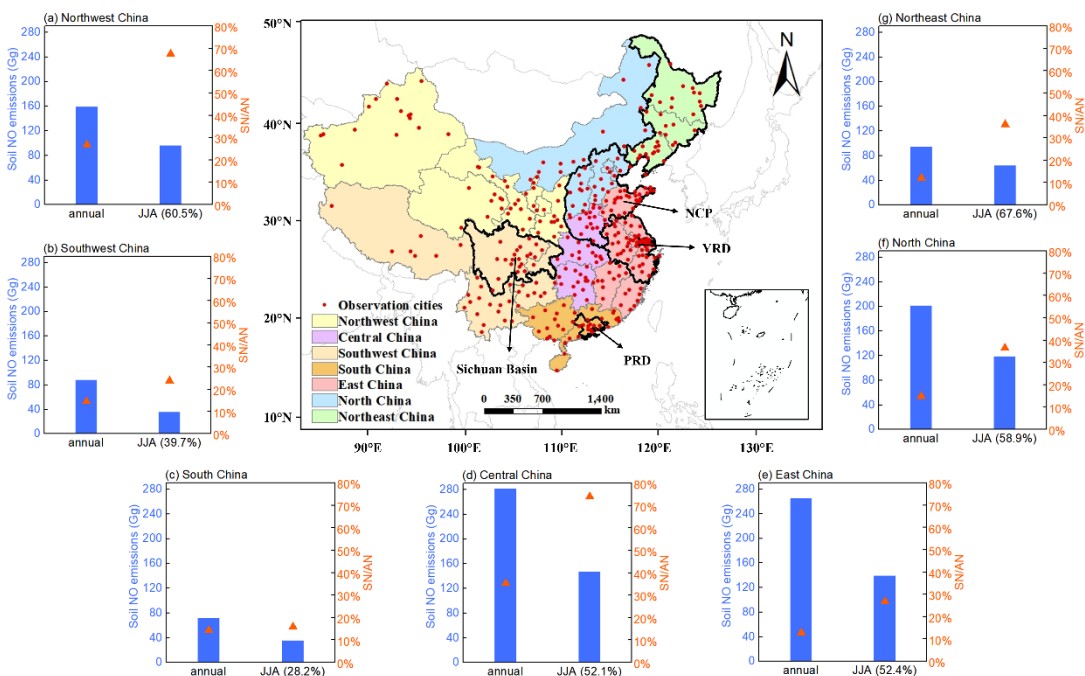


**Figure 1.** Modeling domain and region definitions. Surrounding charts show the annual and
summer (June-July-August, JJA) soil NO emissions and ratio of soil NO to anthropogenic
$NO_x$ emissions for each region.

2.2. Model configurations
A typical modeling approach was applied to evaluate the contribution of soil NO emissions to
surface ozone concentration. The Weather Research and Forecasting (WRF) model (version
3.7, https://www.mmm.ucar.edu/wrf-model-general, accessed on December 1st, 2021) and the
Comprehensive Air Quality Model with Extension (CAMx, version 7.0,
http://www.camx.com/, accessed on December 1st, 2021) were applied to simulate the
meteorological fields and subsequent ozone concentrations. Table S3 listed the detailed model
configurations for WRF and CAMx. Anthropogenic emissions include the Multi-resolution
Emission Inventory of China for 2017 (MEIC, http://www.meicmodel.org, accessed on
December 1st, 2021) and the 2010 European Commission's Emissions Database for Global
Atmospheric Research (EDGAR, http://edgar.jrc.ec.europa.eu/index.php, accessed on
December 1st, 2021) for outside China. Biogenic emissions were calculated along with the
soil NO emissions using MEGAN3.2. Open biomass burning emissions are adopted from the
Fire          INventory          from          NCAR          version          (FINN,          version          1.5,
https://www.acom.ucar.edu/Data/fire/) with MOZART speciation and converted to CAMx
CB05 model species. The gaseous and aerosol modules used in CAMx include the CB05
chemical mechanism (Yarwood et al., 2010) and the CF module. The aqueous-phase
chemistry is based on the updated mechanism of the Regional Acid Deposition Model
(RADM) (Chang et al., 1987). A base case simulation was conducted for June 2018 when soil
NO emissions reached maxima (Section 3.1) and ozone pollution was severe over eastern
China (Mao et al., 2020; Jiang et al., 2022). Base case model performances have been
evaluated in our previous studies (Huang et al., 2021; Huang et al., 2022b). Here we evaluated
simulated ozone concentrations using the Pearson correlation coefficient (R), mean bias
(MB), root-mean-square error (RMSE), normalized mean bias (NMB), and normalized mean
error (NME) against hourly observed ozone concentrations for 365 cities in China. The
formula for each of the statistical metrics is given in Table S4. Observed hourly ozone
concentrations were obtained from the China National Environmental Monitoring Center.
2.3. Brute-force and OSAT
In this study, two methods were used to quantify the impact of soil NO emissions on surface
ozone concentration during the simulation period. The first is the conventional brute-force
method (BFM), which involves comparing the simulated ozone concentration between the
base case and a scenario case without soil NO emissions. The difference between these two
scenarios was considered to represent the contribution of soil NO emissions to ozone. The
second method applies the widely used Ozone Source Apportionment Technology (OSAT)
implemented in CAMx (Yarwood et al., 1996), with soil NO emissions being tagged as an
individual emission group. OSAT attributes ozone formation to $NO_x$ or VOCs based on their
relative availability and apportions $NO_x$ and VOCs emissions by source group/region
(Ramboll, 2021). In addition to soil NO emissions, anthropogenic and natural emissions
(including biogenic VOC emissions, lightning NO emissions, and open biomass burning)
were also tagged as individual emission groups.

**3. Results and discussions**

3.1. Soil NO emissions for 2018 in China

3.1.1. Spatial and temporal variations

National total soil NO emissions for 2018 is estimated to be 1157.9 Gg N, with an uncertainty range of 715.7~1902.6 Gg N, which will be discussed more in Section 3.1.2. On an annual scale, soil NO emissions accounted for 17.3% of the total anthropogenic $NO_x$ emissions in China for 2017 (based on MEIC inventory). This ratio varies from 12.0% to 35.3% at regional scale. Unlike the anthropogenic $NO_x$ emissions that concentrate over densely populated regions (e.g., NCP, YRD), soil NO emissions are most abundant in Central China, particularly Henan Province and nearby provinces, including Hebei and Shandong in the NCP, Jiangsu and Anhui in northern YRD (Fig. 2a). Other hotspots of soil NO emissions include Northeast China and the eastern part of the Sichuan Bain. As expected, the spatial distribution of soil NO emissions closely mirrors that of the fertilizer application (Fig. 2b). Henan (located in Central China), Shandong (NCP), and Hebei (NCP) are the top three provinces that have the highest fertilizer application (together accounting for 24.1% of national totals in 2018) and thus highest soil NO emissions (together accounting for 35.7%).

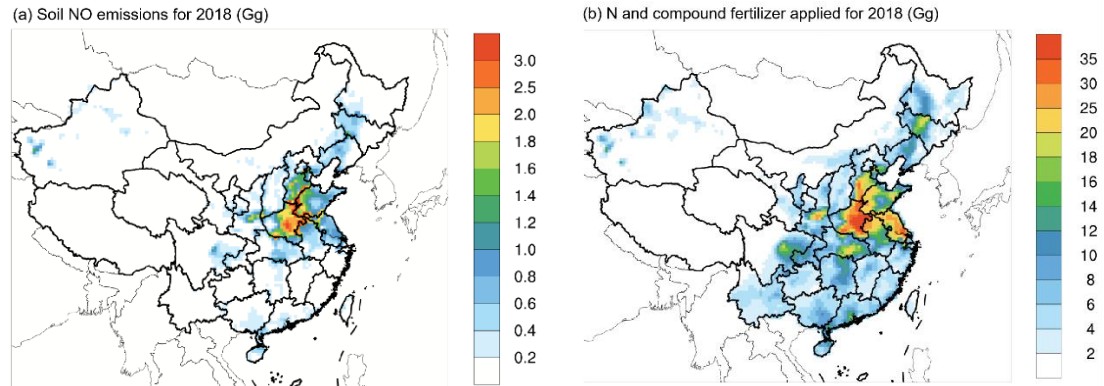

**Figure 2.** Spatial distribution of (a) soil NO emissions for 2018 and (b) N and compound fertilizer applied for 2018.

In terms of the monthly variations, the total soil NO emissions show a unimodal pattern (as shown in Fig. 3a with the highest emissions occurring in the summer months of June, July, and August), except for South China and Southeast China (Fig. S2), where the peak emissions occur in April or May. Soil NO emissions during the summer months account for 28.2% (South China) to 67.6% (Northeast China) of the annual totals (Fig. 1 and Table S5). The shape of monthly soil NO emissions is influenced by temperature and the timing of fertilizer application. The BDSNP algorithm assumes that 75% of the annual fertilizer is applied over the first month of the growing season, with the remaining 25% applied evenly throughout the rest of the growing season. This assumption results in a significant amount of fertilizer being

applied from April to August (Fig. 3a). In contrast, anthropogenic $NO_x$ emissions display
weaker monthly variations (Zheng et al., 2021). Consequently, the ratio of soil NO emissions
to anthropogenic $NO_x$ (SN/AN) is much higher during the summer months. In regions such as
Central China and Northwest China, where soil NO emissions are high and anthropogenic
$NO_x$ emissions are relatively low, SN/AN reaches 74.0% and 67.5% during the summer
months (Fig. 1 and Table S5). In East China and North China, where anthropogenic $NO_x$
emissions are high, SN/AN ranges from 26.8% to 36.5% during the summer months. These
findings are align with Chen et al. (2022), who reported that soil NO emissions made up 28%
of total $NO_x$ (soil NO + anthropogenic $NO_x$) emissions in summer and could reach 50–90% in
isolated areas and suburbs. The substantial contribution of soil NO emissions during the
ozone pollution season implies a potentially significant impact on surface ozone
concentration. In terms of diurnal variations, soil NO emissions peak in the afternoon due to
diurnal temperature fluctuations. As illustrated by Fig. 3b, the average hourly soil NO
emissions over NCP for June 2018 closely follow the WRF simulated temperature changes.
The BDSNP algorithm identifies three sources of soil nitrogen: background, atmospheric
nitrogen deposition, and fertilizer application, with the latter being the primary contributor. A
decomposition analysis of soil NO emissions for NCP reveals that fertilizer application
accounts for 83.4% of total NO soil emissions (Fig. 3b), while background and atmospheric
nitrogen deposition only contribute for 11.2% and 5.4%, respectively. Thus, although soil NO
emissions are generally considered a "natural" source (Galbally et al., 2008) and are not
currently targeted in $NO_x$ emission mitigation strategies, human fertilizer activities render soil
NO emissions an anthropogenic source.

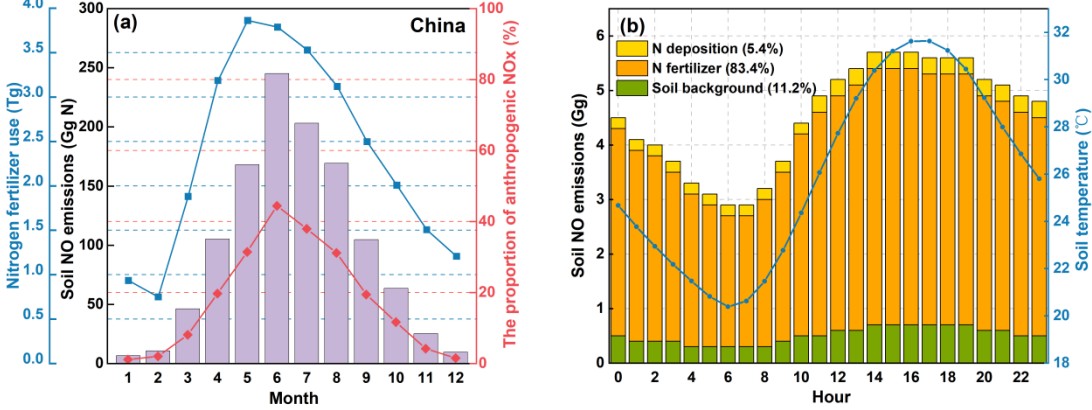

**Figure 3.** (a) Monthly fertilizer (N + compound) applied and soil NO emissions in China and
(b) hourly soil NO emissions for 2018 June in NCP and domain-averaged soil temperature
simulated by WRF.
3.1.2. Limitations and uncertainties associated with soil NO emission estimation
Although the current BDSNP algorithm is considered more sophisticated than the old YL95

algorithm, it still suffers certain limitations. For example, the current BDSNP parameterization employs a static classification of "arid" versus "non-arid" soils, upon which the relationship between soil NO emissions and soil moisture relies (Hudman et al., 2012). However, recent studies (Sha et al., 2021; Huber et al., 2023) have shown more dynamic representation of this classification is needed to capture the emission characteristics as observed by many chamber and atmospheric studies (e.g., Oikawa et al. (2015); Huang et al. (2022a)). Huber et al. (2023) also showed that the emission estimated based on the static classification are very sensitive to the soil moisture and thus could not produce self-consistent results when using different soil moisture products.

In addition to the aforementioned limitation, the estimated soil NO emissions are also subjected to certain limitations and large uncertainties. The first uncertainty comes from the amount of fertilizer application, which has been identified as the dominant contributor to soil NO emissions, as mentioned above. According to the global dataset (Potter et al., 2010), the amount of fertilizer applied is 19.6 Tg, which is comparable to the sum of nitrogen fertilizer for 2018 (20.7 Tg) obtained from provincial statistical yearbooks. However, compound fertilizer, usually with a nitrogen, phosphorus, and potassium ratio of 15: 15: 15, has been used more in China. Each number represents the percentage of the nutrient by weight in the fertilizer. In the case of 15:15:15 NPK fertilizer, it means that the fertilizer contains 15% nitrogen, 15% phosphorus, and 15% potassium. Since 2016, the amount of nitrogen fertilizer has been decreasing annually at an average rate of 4.6%, while the amount of compound fertilizer has been increasing since 2010 at an average rate of 3.3%. The ratio of compound fertilizer to nitrogen fertilizer has increased from 76.4% in 2010 to 109.8% in 2018. Consequently, soil NO emissions may be largely underestimated if the compound fertilizer is not taken into account. Our calculation shows that if only nitrogen fertilizer is considered, the estimated total soil NO emissions are 805.2 Gg N/a for 2018, which is comparable to the value (770 Gg N/a averaged during 2008-2017) reported by Lu et al. (2021), but 30.5% lower than that based on both nitrogen fertilizer and compound fertilizer. Regionally, this underestimation ranges from 11.1%~41.5%, with a larger underestimation in Central China and East China (Fig. S3).

Another major uncertainty in estimating soil NO emissions is the temperature dependence factor $f(T)$ in Eq.1. According to the BDSNP scheme, soil NO emissions increase exponentially with temperature between 0 and 30°C and reach a maximum when the temperature exceeds 30°C. The default temperature dependence coefficient (i.e., $k$ in Eq. S2) follows the value used in the YL95 scheme, which is 0.103±0.04. However, as shown by Table 3 in Yienger and Levy (1995), this value is the weighted average of values reported for different land types, which shows a wide range from 0.040 to 0.189. Even for the same crop type (e.g., corn), the value of $k$ could be quite different (0.130 vs. 0.066). We conducted a

sensitivity analysis to examine the impact of varying the $k$ value on estimated soil NO emissions. When the $k$ value decreases or increases by 20%, the estimated total soil NO emissions change from 715.7 to 1902.6 Gg N/a, representing a relative difference of -38.2~64.3% deviation from the default value (1157.9 Gg N/a). Using the default $k$ value would result in a large overestimation of simulated $NO_2$ concentrations over NCP and YRD and underestimation over Northeast China (Fig. S4). According to the total sown areas of farm crops reported in the provincial statistical yearbook, the primary crops grown in these regions are wheat and corn, which have a relatively low $k$ value (0.066~0.073). Therefore, we adjusted $k$ for NCP (reduced by 20%), YRD (reduced by 10%), and Northeast China (increased by 10%). CAMx simulation results show that this adjustment would not significantly affect the simulated MDA8 $O_3$ concentration but could reduce the $NO_2$ gap between observation and simulation (Fig. S4-S5). Therefore, we applied this adjustment to soil NO emissions in the following CAMx simulations.

3.2. Contribution of soil NO emissions to ground-level ozone

3.2.1. Base case model evaluation

Fig. 4 shows the monthly averaged MDA8 ozone concentration simulated for June 2018 with observed values presented on top. Overall the model well captured the spatial distribution of MDA8 with a spatial correlation R = 0.89. Over the 365 cities in China, the simulated monthly averaged MDA8 ozone concentration is 146.7±36.1 μg/m$^3$, which is slightly higher than the observed value of 129.6±37.6 μg/m$^3$ (NMB = 13.2%). Regionally, model shows better performance in Northeast China (MB = 2.4 μg/m$^3$, NMB = 1.9%) and NCP (MB = 13.3 μg/m$^3$, NMB = 7.7%). Over-prediction is observed for Sichuan Basin and YRD (Table S6). Simulated ozone concentration over the northwest Qinghai-Tibet Plateau was also much higher than observed values. Our OSAT results (shown later) show that the high ozone concentration over the Qinghai-Tibet Plateau is mostly contributed by the transport of boundary ozone, which includes both horizontal and vertical (i.e., stratosphere) directions. For regions with high altitude (e.g., the Qinghai-Tibet Plateau), vertical ozone intrusion from the stratosphere is most substantial, which is consistent with the finding by Chen et al. (2023) that the boundary layer height was identified as the most important feature for ozone over the Qinghai-Tibet Plateau.

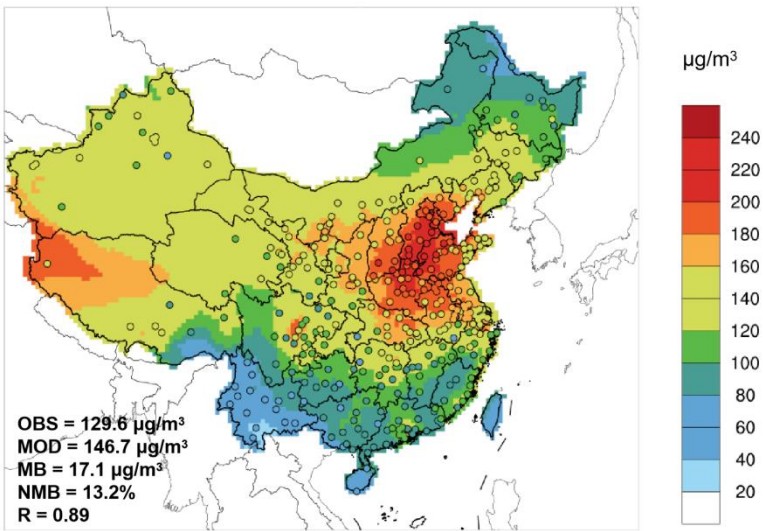

**Figure 4.** Comparison of simulated (base colors) and observed (scatter points) values of MDA8 ozone in June 2018.

3.2.2. Impacts on regional ozone

To assess the contribution of soil NO emissions to surface ozone, both the brute-force method (BFM) and the OSAT method were applied, and the results are shown in Fig. 5. Generally, the two methods show consistent ozone contribution from soil NO emissions but with different magnitudes. The BFM method shows widespread ozone enhancement due to soil NO emissions with a spatial pattern that aligns with the distribution of soil NO emissions. Substantial ozone enhancement is found over Central China, Sichuan Basin, northern YRD, and eastern Northeast China. Maximum ozone enhancement ($\Delta$MDA8) due to soil NO emissions is 26.4 $\mu g/m^3$ with a domain-average value of 8.0 $\mu g/m^3$. For selected key regions, the ozone contribution ranges from low to high: PRD (3.8$\pm$1.1 $\mu g/m^3$), YRD (8.7$\pm$4.7 $\mu g/m^3$), Sichuan Basin (9.1$\pm$0.9 $\mu g/m^3$), Northeast (9.3$\pm$3.0 $\mu g/m^3$), and NCP (13.9$\pm$4.4 $\mu g/m^3$), respectively. A similar spatial pattern is observed with the OSAT results, but the magnitudes are much higher. Maximum ozone contribution by soil NO emissions reaches 40.4 $\mu g/m^3$ according to OSAT results, which is 53.0% higher than the brute force method. The corresponding ozone contribution for each selected region is 6.7$\pm$1.2 $\mu g/m^3$ (PRD), 13.5 $\pm$7.4 $\mu g/m^3$ (Sichuan Basin), 14.5$\pm$4.9 $\mu g/m^3$ (Northeast China), 16.2$\pm$7.8 $\mu g/m^3$ (YRD) and 25.7$\pm$5.3 $\mu g/m^3$ (NCP). The scatter plots between BFM and OSAT results show good correlations (Fig. S6, $R^2$ = 0.78~0.97), with OSAT results higher by 10%~61%. For YRD, Sichuan Basin, and Northeast, the difference between the OSAT method and BFM increases with the absolute ozone concentration (Fig. S7), while NCP shows the opposite trend. The difference between the two methods reflects the nonlinear ozone response to $NO_x$ emissions. This nonlinearity becomes stronger in regions with larger $NO_x$ concentrations, especially where $O_3$ production is characterized as $NO_x$-saturated (or VOC-limited), such as the NCP. In

such cases, removing a portion of the NO emissions (e.g., zeroing out soil NO for the BFM simulation) makes $O_3$ production from the remaining NO emissions more efficient, which lessens the $O_3$ response. As shown later in Figure 7a, the $O_3$ response for NCP is more curved (nonlinear) than other regions, consistent with NCP tending to have more $NO_x$-saturated $O_3$ production. This nonlinear effect also explains smaller $O_3$ attribution to soil NO by the BFM than OSAT, especially over the NCP. Attributing a secondary pollutant to a primary emission (e.g., $O_3$ to NO) is inherently tricky with nonlinear chemistry, as Koo et al. (2009) discussed. Therefore, it is useful to present estimates from different methods. The Path Integral Method (PIM) is a source apportionment method that explicitly treats nonlinear responses with mathematical rigor (Dunker et al., 2015). However, applying the PIM is more costly than the BFM or OSAT.

In addition to soil NO contribution, OSAT also gives ozone contributions from other source groups, including anthropogenic emissions within China, boundary contribution, natural emissions (e.g., biogenic emissions, open biomass burning, lightning $NO_x$), and emissions outside China. The spatial distribution for each source category is presented in Fig. S8, and the relative contribution for each selected region is shown in Fig. S9. Overall, boundary transport (56.5%) and anthropogenic emissions (24.0%) contribute most to MDA8 ozone for June 2018. Boundary contribution is high over the western and northern parts of China, while the contribution from anthropogenic emissions is substantial over eastern China, where anthropogenic emissions are extensive. On a national scale, soil NO emissions exhibit a relative ozone contribution of 9.1%, and regionally this value ranges from 6.1% in PRD to 13.8% in NCP.

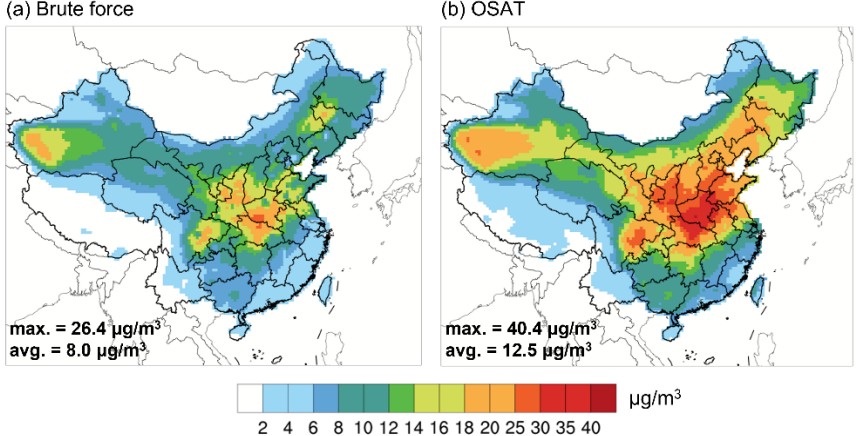

**Figure 5.** Ozone contribution from soil NO emissions based on (a) brute force method and (b) OSAT method.

We further evaluated the impact of soil NO emissions on the number of ozone exceedances days (i.e., days with MDA8 $O_3$ higher than 160 $\mu g/m^3$) during June 2018 based on the relative response factor (RRF) method and results from the brute force method. The total number of

ozone exceedances days during June 2018 for the five selected regions ranged from 50 days in PRD to 985 days in NCP (Table 1). The number of ozone exceedance days per city ranged from 3.1 days in Sichuan Basin to 18.2 days in NCP, suggesting the severe ozone pollution in June 2018 over NCP. RRF was first calculated for each city as the ratio of simulated ozone concentration between the base case and the case with soil NO emissions excluded and applied to the observed ozone concentrations to obtain adjusted ozone concentrations without soil NO emissions. Soil NO emissions are estimated to lead to 121 ozone exceedance days in NCP, followed by 84 days in the Northeast and 70 days in YRD, corresponding to a percent change of 12.3%, 32.8%, and 10.5%, respectively. In Sichuan Basin, where soil NO emissions are also substantial, soil NO emissions contribute 30 ozone exceedances days, which accounts for 43.5% of the total ozone exceedances days. These results suggest the substantial contribution of soil NO emissions to the number of ozone pollution days over regions with high soil NO emissions.

**Table 1.** Number of ozone exceedances over selected regions during June 2018.

| Region (No. of cities) | Number of ozone exceedance days (% of total days) | Δozone exceedances days when soil NO emissions are removed | % of total ozone exceedances days |
|---|---|---|---|
| NCP (54) | 985 (60.8%) | -121 | -12.3% |
| YRD (55) | 666 (41.1%) | -70 | -10.5% |
| PRD (9) | 50 (18.5%) | -6 | -12.0% |
| Sichuan Basin (22) | 69 (10.5%) | -30 | -43.5% |
| Northeast (37) | 256 (23.1%) | -84 | -32.8% |

3.3. Ozone responses to reductions in soil NO emissions

Current $NO_x$ emission control policies primarily target combustion sources, such as power plants (Du et al., 2021) and on-road vehicles (Park et al., 2021). Nitrification inhibitors, such as dicyandiamide (DCD, $C_2H_4N_4$), have been found to be effective in reducing nitrogen loss, thereby reducing NO emissions from soil (Abalos et al., 2014). Studies have shown that using 5% DCD with nitrogen fertilizer can reduce NO emissions by up to 70% (Xue et al., 2022). In light of this, it is important to evaluate the impact of reduced soil NO emissions on ozone concentration. To address this question, four sensitivity simulations were carried out for June 2018, with soil NO emissions reduced by 25%, 50%, 75%, and 100% relative to the base case. As shown by Fig. 6, reducing soil NO emissions led to a general decrease in monthly MDA8 ozone concentration (ΔMDA8), with the magnitude of ΔMDA8 becoming more significant with the reduction ratio. With a 25% reduction in soil NO emissions, there was a widespread small decrease in monthly average MDA8 ozone concentration (ΔMDA8: -1.5±0.9 $\mu g/m^3$), except over NCP where ozone showed a slight increase (up to 1.3 $\mu g/m^3$) in Shandong and Henan province. These ozone increases reflect the nonlinearity of ozone

chemistry and this nonlinearity becomes stronger in regions with large $NO_x$ concentrations, especially where $O_3$ production is characterized as VOC-limited (such as NCP). When soil NO emissions were cut by 50%, the effect of reduced $O_3$ titration is overwhelmed by reduced $O_3$ formation due to less $NO_x$ available, thus the ΔMDA8 showed a ubiquitous decrease across entire China with an average ΔMDA8 of -5.5 μg/m³. When soil NO emissions were removed entirely, the maximum ΔMDA8 could exceed 25 μg/m³ over central China, part of the Sichuan Basin, Northeast China, and Northeast China. Regions with strong ozone responses generally aligned with regions that also had high soil NO emissions. However, it should be noted that the ozone response to soil NO reductions not only depends on the magnitude of soil NO emissions but is also affected by (1) the local ozone formation regime that is further determined by the relative abundance of $NO_x$ and VOCs, and (2) changes in transport of upwind ozone.

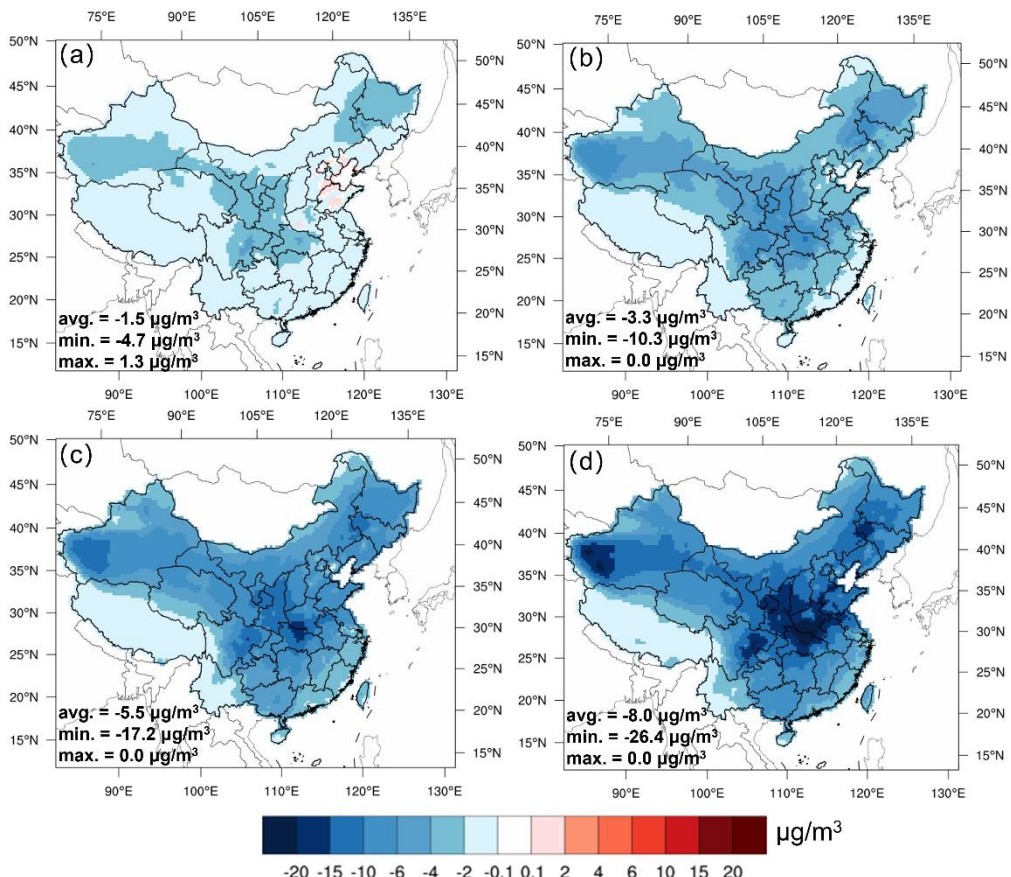

**Figure 6.** Spatial distribution of ΔMDA8 under (a) 25%, (b) 50%, (c) 75%, and (d) 100% reductions of soil NO emissions in June 2018.

Fig. 7a provides further details on the domain-averaged ΔMDA8 under different reduction scenarios for the five key regions. As expected, the ozone response in each region increased as the reduction in the soil NO emissions increased. NCP exhibited the strongest ozone responses to changes in soil NO emissions, with ΔMDA8 increasing from -0.7±0.8 μg/m³ with 25% reductions

to -13.9±4.4 μg/m³ when all soil NO emissions were removed. YRD, Sichuan Basin, and Northeast China exhibit similar ozone responses when soil NO emissions are reduced. Under the 25% scenario, ΔMDA8 ranged from -4.7 to 1.3 μg/m³ for these three regions; with 100% soil NO reductions, ΔMDA8 ranged from -21.4 to -0.9 μg/m³. ΔMDA8 in PRD was relatively small. Even with a 100% reduction, the average ΔMDA8 in PRD was less than 5 μg/m³, which is associated with the small soil NO emissions in PRD. It is interesting to note that all regions except NCP exhibited an approximate linear ozone response to changes in soil NO emission reductions. NCP showed more significant ozone reductions as the reduction ratio increased, suggesting that NCP would gain more benefits with more aggressive reductions in soil NO emissions compared to other regions.

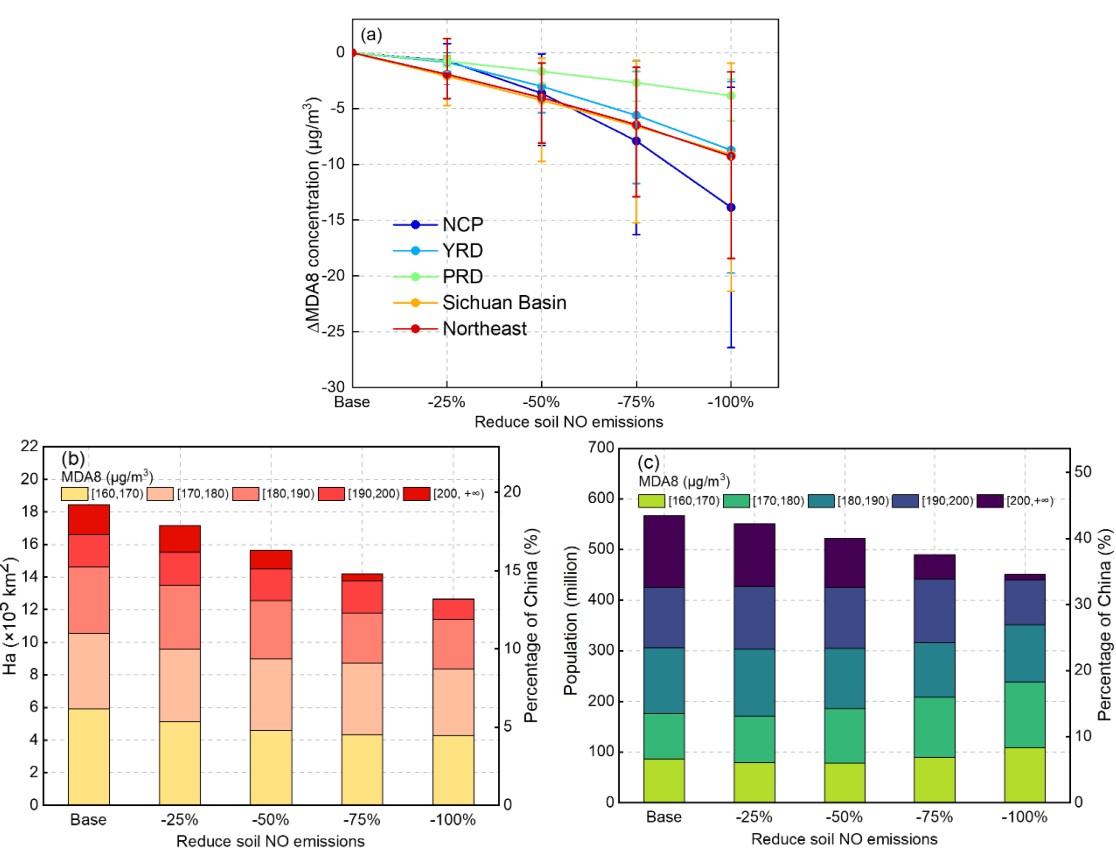

**Figure 7.** (a) ΔMDA8 concentrations in five key regions under different emission reduction scenarios (b) Area and (c) population exposed to different ozone levels under different soil NO emission reduction scenarios.

We evaluated the impact of different soil NO emission reduction scenarios on the area and population exposed to varying ozone levels. The results, presented in Fig. 7b and 7c, revealed a decrease in coverage and exposed population under high ozone concentrations as soil NO emissions decrease. The data presented in the plots are for grid cells with monthly MDA8 ozone concentrations exceeding 160 μg/m³. In the Base scenario, the estimated coverage of MDA8 ozone exceeding 160 μg/m³ was 1.84×10⁶ km², equivalent to 19.2% of the national

land area. The population exposed to ozone concentrations exceeding 160 μg/m$^3$ amounts to
566.6 million, representing 43.4% of the entire population. The areas with extremely high
ozone concentrations (MDA8 > 200 μg/m$^3$) account for 1.9% of the national land area, with a
corresponding exposed population of 10.9%, indicating that densely populated areas
experience higher ozone concentrations. When soil NO emissions are halved, there is a 15.2%
reduction in the coverage of non-attainment areas and an 8.0% reduction in the total exposed
population. If soil NO emissions are eliminated, the total area coverage and population
exposed to MDA8 ozone concentrations exceeding 160 μg/m$^3$ would be $1.27 \times 10^6$ km$^2$ and
450.3 million, respectively, representing 13.2% and 34.5% of the total. Compared to the Base
scenario, a 100% theoretical reduction in soil NO emissions leads to a 31.3% and 20.5%
reduction in the exposed area and population under high ozone concentration, respectively,
indicating substantial health benefits gained when soil NO emissions are mitigated.
Fig. S10-S11 displays similar area and population plots for selected key regions. The overall
trends for each sub-region are consistent. With 100% reductions in soil NO emissions, the
area with high ozone concentration decreased by 17.8%, 22.3%, 65.4%, and 100% for NCP,
YRD, Sichuan Basin, and Northeast. The corresponding values for the exposed population are
91.4%, 60.3%, 9.8%, and 0.0%. While the relative change is more significant in Sichuan
Basin and Northeast China, NCP and YRD gain more health benefits due to the significantly
higher total population for these two regions. However, it is worth noting that even with the
complete elimination of soil NO emissions, a total of 450.3 million people are still exposed to
ozone levels exceeding the national standard, necessitating multiple control policies at the
same time, such as synergistic control of anthropogenic VOC emissions (Chen et al., 2022;
Ding et al., 2021).
3.4 Comparison with existing studies
The soil NO emissions estimated in this study were also compared with values reported by
existing studies based on either field measurement or model estimation (Table S7). Previous
studies report a wide range of soil NO emissions from 480 to 1375 Gg N and soil NO flux
ranging from 10 to 47.5 ng N m$^{-2}$ s$^{-1}$. The soil NO emissions estimated in our study are 1157.9
Gg N with the default $k$ value and 951.9 Gg N with region-adjusted $k$ value, which falls
within the upper range of previously reported values. The averaged soil NO flux over NCP in
June 2018 estimated in our study is 35.4 ng N m$^{-2}$ s$^{-1}$, which is within the range reported by
previous studies (12.9~40.0 ng N m$^{-2}$ s$^{-1}$).
The simulated ozone contribution by soil NO emissions is compared with other studies. In
California, soil NO was estimated to cause a 23.0% increase in surface O$_3$ concentrations (Sha
et al., 2021). Constrained by satellite measured NO$_2$ column densities, Wang et al. (2022b)
reported MDA8 ozone contribution of 9.0 μg/m$^3$ (relative contribution of 5.4%) from

cropland NO$_x$ emissions over NCP during a growing season in 2020. Lu et al. (2021) showed an interactional effect of domestic anthropogenic emissions with soil NO emissions of 9.5 ppb in the NCP during July 2017. In addition, soil NO$_x$ emissions strongly affect the sensitivity of ozone concentrations to anthropogenic sources in the NCP. In a most recent study by Shen et al. (2023), addition of the soil NO$_x$ emissions was shown to result in up to 15 ppb increase of ozone concentration over Xinjiang, Tibet, Inner Mongolia, and Heilongjiang, although a minor reduction was evident over the Yangtze River basin. The findings of this study align with previous studies, emphasizing the important role of soil NO emissions in influencing surface ozone concentrations in China. Furthermore, spatial heterogeneities exist in terms of both the soil NO emissions and the responses of ozone to reductions in soil NO emissions. However, it should be noted that the spatial pattern of ozone response to reduced soil NO emissions in this study is different from Shen et al. (2023). For instance, with a 30% reduction in soil NO emissions, O$_3$ concentration increased by 3-5 ppb over Inner Mongolia, Heilongjiang, Xinjiang, and Tibet and decreased by 0-2 ppb over the Yangtze River basin in Shen et al. (2023). In this study, a 20% reduction in soil NO emissions was found to lead to widespread but small decrease (less than 4 µg/m$^3$) in ozone concentrations except the NCP (Fig. 6a). These inconsistences may stem from the differences in the estimated soil NO emissions, both associated with the magnitude and the spatial distribution, as also noted in other study (Zhu et al., 2023). Therefore, more observations, such as direct measurement of soil NO flux, especially over agricultural areas, are urgently needed to better constrain the estimated soil NO emissions.

**4. Conclusions**

Soil NO emissions are non-negligible NO$_x$ sources, particularly during summer. The importance of soil NO emissions to ground-level ozone in China is much less evaluated than combustion NO$_x$ emissions. In this study, the total national soil NO emissions were estimated to be 1157.9 Gg N in 2018 based the BDSNP algorithm, with a spatial distribution closely following that of fertilizer application. High soil NO emissions were greatest over Henan, Shandong, and Hebei provinces, which differs significantly from where anthropogenic NO$_x$ emissions are. Distinct diurnal and seasonal variations in soil NO emissions were found, mainly driven by the changes in soil temperature as well as the timing of fertilizer application. Uncertainty analysis of the estimated soil NO emissions reveals a range of 715.7~1902.6 Gg N that warrants further study and, preferably, constraint from observations.

Using two ozone source attribution methods (BFM and OSAT), we evaluated the contribution of soil NO emissions to ground-level ozone concentration for June 2018. Both methods suggest a substantial contribution of soil NO emissions to MDA8 ozone concentrations of 8~12.5 µg/m$^3$ on average for June 2018, with the OSAT results consistently higher than BFM.

Soil NO emissions were shown to increase of ozone exceedances days (i.e., MDA8 above 160 $\mu g/m^3$) by 10.0%~43.5% depending on region. Reducing soil NO emissions could generally reduce the ground-level ozone concentrations and population exposure to unhealthy ozone levels, especially over NCP and YRD. For example, a 50% reduction in soil NO emissions decreased land area experiencing ozone above 160 $\mu g/m^3$ by 15.2% and the population exposed to this ozone concentration by 8.0%. However, even with complete removal of soil NO emissions, approximately 450.3 million people are still exposed to ozone above 160 $\mu g/m^3$.

The major findings of this study reinforce previous studies by highlighting the important contribution of soil NO emissions to surface ozone concentrations in China, although substantial uncertainties remain with soil NO emission estimates. Observational constraints on the magnitude of soil $NO_x$ emissions in China are needed. Ozone response to reducing soil NO emissions varies by region due to the non-linear chemistry of ozone formation. Future ozone mitigation strategies should consider the potential benefit of reducing non-combustion $NO_x$ emissions, such as soil NO, with due consideration to the sensitivity of ozone to reducing $NO_x$ in the region.

**Data availability.** Data will be made available on request.

**Author contributions. Ling Huang:** Conceptualization, Formal analysis, Writing – original draft. **Jiong Fang:** Data curation, Formal analysis, Visualization. **Jiaqiang Liao:** Data curation, Formal analysis, Visualization. **Greg Yarwood:** Writing – review & editing. **Hui Chen:** Writing – review & editing. **Yangjun Wang:** Writing – review & editing. **Li Li:** Conceptualization, Supervision, Funding acquisition.

**Competing interests.** The authors declare that they have no known competing financial interests or personal relationships that could have appeared to influence the work reported in this paper.

**Acknowledgments.** This study was financially sponsored by the National Natural Science Foundation of China (grant No. 42005112, 42075144), the Open Funding of Zhejiang Key Laboratory of Ecological and Environmental Big Data (No. EEBD-2022-06), the Shanghai International Science and Technology Cooperation Fund (No.19230742500). This work is supported by Shanghai Technical Service Center of Science and Engineering Computing, Shanghai University.

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
