# Peer review of "Insights into Soil NO Emissions and the Contribution to Surface Ozone Formation in China"

_EGUsphere, 2023_

## Author Comment (AC1)

**Response to reviewers' comments**

**Reviewer #1:**

In this paper, the authors used two methods to quantify the effect of soil NO emissions on surface ozone concentrations during the simulation period. The first is the traditional approach (BFM), which involves comparing simulated ozone concentrations in the base case with scenarios without soil NO emissions. The difference between the two scenarios is thought to represent the contribution of soil NO emissions to ozone. The second method uses the widely used ozone source allocation technique (OSAT) implemented by CAMx to label soil NO emissions as separate emission groups. The two methods basically reached the same conclusion: soil NO emission has a great effect on ozone concentration. This study emphasizes the importance of considering soil NO emissions in future ozone mitigation strategies in China. The article is integrative and innovative, and can be published. But it also has the following problems that need to be addressed by the authors.

Response: The authors thank the reviewer for reading the manuscript carefully and providing valuable comments. Our point-to-point response is given below. Revisions made to the manuscript and Supporting Information are highlighted in yellow.

**Major question:**

1. The Manuscript needs to add a discussion section in which the inclusion of comparisons with other models and the provision of comparative results with other relevant studies or models can provide a broader context for the application of the findings. More detailed information on the evaluation results and potential uncertainties in the model simulations would improve the robustness of the analysis.

Response: Thanks for the comment. We added a new section "3.4 Comparisons with existing studies" in the revised manuscript to specifically compare our estimated soil NO emissions as well as simulated results with existing studies (L450-L473):

**"3.4 Comparison with existing studies"**

The soil NO emissions estimated in this study were also compared with values reported by existing studies based on either field measurement or model estimation (Table S7). Previous studies report a wide range of soil NO emissions from 480 to 1375 Gg N and soil NO flux ranging from 10 to 47.5 ng N m-2 s-1. The soil NO emissions estimated in our study are 1157.9 Gg N with the default k value and 951.9 Gg N with region-adjusted k value, which falls within the upper range of previously reported values. The averaged soil NO flux over NCP in June 2018 estimated in our

study is 35.4 ng N m-2 s-1, which is within the range reported by previous studies  $(12.9 \sim 40.0 \text{ ng N m}^{-2} \text{ s}^{-1})$ .

The simulated ozone contribution by soil NO emissions is compared with other studies. In California, soil NO was estimated to cause a 23.0% increase in surface  $O_3$ concentrations (Sha et al., 2021). Constrained by satellite measured NO2 column densities, Wang et al. (2022b) reported MDA8 ozone contribution of 9.0  $\mu g/m^3$ (relative contribution of 5.4%) from cropland NOx emissions over NCP during a growing season in 2020. Lu et al. (2021) showed an interactional effect of domestic anthropogenic emissions with soil NO emissions of 9.5 ppb in the NCP during July 2017. In addition, soil NOx emissions strongly affect the sensitivity of ozone concentrations to anthropogenic sources in the NCP. In a most recent study by Shen et al. (2023), addition of the soil NOx emissions was shown to result in up to 15 ppb increase of ozone concentration over Xinjiang, Tibet, Inner Mongolia, and Heilongjiang, although a minor reduction was evident over the Yangtze River basin. When soil NOx emissions were reduced by 30%, ozone concentrations increased by 3-5 ppb over Inner Mongolia, Heilongjiang, Xinjiang, and Tibet, while decreased by 0-2 ppb over the Yangtze River basin. Surprisingly, when soil NOx emissions were increased by 30%, nearly identical ozone responses were observed."

(2) We also performed a detailed uncertainty analysis of the estimated soil NO emissions in "Section 3.1.2 Limitations and Uncertainties associated with soil NO emission estimation". We considered two factors that might cause uncertainties in estimating the soil NO emissions: the amount of fertilizer application and the temperature dependence factor ( $\beta$ ) used in soil NO calculation. We also added discussions on the limitation of the current BDSNP algorithm related to the static classification of "arid" versus "non-arid", as suggested by the other reviewer (L244-L292):

[revised manuscript text omitted]

| Reference         | Region | Reference
year | Above
canopy
(Gg N a -1 ) | Soil NOx flux (ng N m -2 s -1 ) | Notes                                      |
|-------------------|--------|-------------------|--------------------------------------------|-------------------------------------------------------|--------------------------------------------|
| Wang et al.       | China  | 1999              | 657                                        | Generally more than 40 ng N m $^{-2}$                 | An empirical modeling approach of Yienger  |
| (2005)            |        |                   |                                            | $s^{-1}$ (in the North China Plain in                 | and Levy (YL95)                            |
|                   |        |                   |                                            | July) and 20 ng N $m^{-2} s^{-1}$ (in the             |                                            |
|                   |        |                   |                                            | northeast China in July)                              |                                            |
| Tie et al. (2006) | China  | 2004              | 1375                                       |                                                       | Dynamical and biogenic emissions models,   |
|                   |        |                   |                                            |                                                       | soil emissions parameterized with an       |
|                   |        |                   |                                            |                                                       | exponential dependence on soil             |
|                   |        |                   |                                            |                                                       | temperature.                               |
| Yan et al. (2005) | China  |                   | 480                                        |                                                       | Statistical model based on field           |
|                   |        |                   |                                            |                                                       | measurements of NOx fluxes combined        |
|                   |        |                   |                                            |                                                       | with land cover, soil pH, soil organic     |
|                   |        |                   |                                            |                                                       | carbon, climate, and nitrogen fertilizers  |
| Huang and Li      | China  |                   | 1226 (ranging                              |                                                       | Synthesis of 130 NO emissions sampling     |
| (2014)            |        |                   | from 588.24 to                             |                                                       | points at 14 locations to estimate soil NO |
|                   |        |                   | 2132.05)                                   |                                                       | emissions inventory in China.              |
| Lu et al. (2019)  | China  | 2016              | 1140                                       |                                                       | BDSNP scheme in GEOS-Chem                  |
|                   |        | (Mar-Oct)         |                                            |                                                       |                                            |

|                   |             | 2017      | 1360            |                                      |                                                    |
|-------------------|-------------|-----------|-----------------|--------------------------------------|----------------------------------------------------|
|                   |             | (Mar-Oct) |                 |                                      |                                                    |
| Lu et al. (2021)  | China       | 2017      | 770 ± 40 |                                      | BDSNP scheme in GEOS-Chem                          |
| Wang et al.       | East China  | 1997-1999 | 850             |                                      | Application of YL95 scheme in                      |
| (2007)            |             |           |                 |                                      | GEOS-Chem                                          |
| Lin (2012)        | East China  | 2006      | 380             |                                      | Top-down estimates using satellite NO 2 |
|                   |             |           |                 |                                      | retrievals                                         |
| (Wang et al.,     | The North   | 2020      |                 | 10-40 (crop growing season)          | Application of MEGAN scheme in                     |
| 2022b)            | China plain |           |                 |                                      | WRF-Chem                                           |
| Li and Wang       | the Pearl   | 2005      |                 | Typical vegetable plot average: 47.5 | NO flux measured by static chamber                 |
| (2007)            | River Delta |           |                 |                                      | technique in the suburbs of Guangzhou              |
| Li et al. (2007)  | South China | 2005      |                 | The average fluxes of broadleaved    | Sample plots circled in the forest and NO          |
|                   |             |           |                 | forest and pine-leaved forest in the | fluxes measured by dynamic flow chamber            |
|                   |             |           |                 | rainy season were 14.9 and 17.1      | technique                                          |
| Liu et al. (2011) | Northern    | 2007-2009 |                 | Average annual flux: 7.6             | Experiments NO fluxes were obtained                |
|                   | China       |           |                 | (wheat-maize rural)                  | based on automatic measurement systems             |
|                   |             |           |                 |                                      | and intermittent manual measurements               |
| Liu et al. (2017) |             |           |                 | Average soil NO flux: 12.9           | Synthesized 520 field observations from            |
|                   |             |           |                 | Vegetable farmland flux: 30.9        | 114 publications                                   |
| This study        | China       | 2018      | 805.2           | 6.6 (June average, same below)       | BDSNP scheme (default fertilizer data)             |
|                   |             |           |                 |                                      |                                                    |

|     |      | 1157.9        | 9.9  | BDSNP scheme (N + compound fertilizer  |
|-----|------|---------------|------|----------------------------------------|
|     |      | (715.7-1902.6 |      | data)                                  |
|     |      | )             |      |                                        |
| NCP | 2018 | 296.1         | 38.5 | BDSNP scheme (default fertilizer data) |
|     |      | 455.9         | 60.1 | BDSNP scheme (N + compound fertilizer  |
|     |      | (276.5-762.1) |      | data)                                  |
|     |      |               | 35.4 | BDSNP scheme (N + compound fertilizer  |
|     |      |               |      | data and adjusted $\beta$ value)       |

2. For the simplicity of the article, some methods can be described briefly, but the main methods involved in the article need to be described carefully to enhance the readability of the article. The paper lacks the detailed introduction of WRF model configuration and model parameterization scheme. In addition, did the author consider the influence of regional differences in precipitation between north and south China on the model when selecting the parameterization scheme?

Response: Thank you for your suggestion. We listed the detailed model configurations in Table S3. We did not consider the influences of regional differences in precipitation when selecting the parameterization scheme since WRF does not permit different parameterization schemes within a single simulation.

| Items                         | WRF                                   | CAMx                            |  |
|-------------------------------|---------------------------------------|---------------------------------|--|
| Version                       | 3.7                                   | 7.0                             |  |
|                               |                                       | Carbon Bond chemistry           |  |
| Gas-Phase chemistry           |                                       | (CB06) (Yarwood et al.,         |  |
|                               |                                       | 2010)                           |  |
| Aerosol module                |                                       | CF Scheme / ISORROPIA           |  |
|                               |                                       | Updated mechanism of the        |  |
| A queque phose chemistry      |                                       | Regional Acid Deposition |  |
| Aqueous-phase chemistry       |                                       | Model (RADM) (Chang et          |  |
|                               |                                       | al., 1987)                      |  |
| Planat boundary sahama        | The Yonsei University Scheme (YSU)    |                                 |  |
| Flanet boundary scheme        | (Hong et al., 2006)                   |                                 |  |
| Land surface                  | NOAH (Ek et al., 2003)                |                                 |  |
| Microphysics                  | Morrison double-moment scheme         |                                 |  |
| Merophysics                   | (Morrison et al., 2009)               |                                 |  |
| Shortwave radiation           | RRTMG shortwave (Iacono et al., 2008) |                                 |  |
| Longwave radiation            | RRTMG scheme (Iacono et al., 2008)    |                                 |  |
| Boundary conditions / initial | 1°×1° grids FNL Operational Global    | MOZART (Emmons et al.,          |  |
| conditions                    | Analysis data archived at the GDAS    | 2010)                           |  |
| Dry deposition and wet        |                                       | Then $\alpha$ et al. (2002)     |  |
| deposition                    |                                       | Zhang et al. (2005)             |  |

 Table S3. Model configurations of WRF and CAMx.

**3. Please explain why the NO value simulated by OSAT model is higher than BFM**

Response: We understand the reviewer is asking about the contribution of soil NO to  $O_3$ , as shown in Fig. 5. Differences in  $O_3$  contribution estimates from OSAT and BFM arise from photochemical nonlinearity in the relationship between  $O_3$  and NO emissions, as seen, for example, in Fig. 7a. This nonlinearity becomes stronger in regions with larger NOx concentrations, especially where  $O_3$  production is

characterized as NO*x*-saturated (or VOC-limited), such as the NCP. In such cases, removing a portion of the NO emissions (e.g., zeroing out soil NO for the BFM simulation) makes  $O_3$  production from the remaining NO emissions more efficient, which lessens the  $O_3$  response. Note that in Fig. 7a, the  $O_3$  response for NCP is more curved (nonlinear) than other regions, consistent with NCP tending to have more NO*x*-saturated  $O_3$  production. This nonlinear effect also explains smaller  $O_3$  attribution to soil NO by the BFM than OSAT, especially over the NCP, as seen in Fig. 5. Attributing a secondary pollutant to a primary emission (e.g.,  $O_3$  to NO) is inherently tricky with nonlinear chemistry, as Koo et al. (2009) discussed, so it is useful to present estimates from different methods. The Path Integral Method (PIM) is a source apportionment method that explicitly treats nonlinear responses with mathematical rigor (Dunker et al., 2015). However, applying the PIM is more costly than the BFM or OSAT. We have added some discussions in the revised manuscript (L332-L346):

"The difference between the two methods reflects the nonlinear ozone response to NOx emissions. This nonlinearity becomes stronger in regions with larger NOx concentrations, especially where O3 production is characterized as NOx-saturated (or VOC-limited), such as the NCP. In such cases, removing a portion of the NO emissions (e.g., zeroing out soil NO for the BFM simulation) makes O3 production from the remaining NO emissions more efficient, which lessens the O3 response. As shown later in Fig. 7a, the O3 response for NCP is more curved (nonlinear) than other regions, consistent with NCP tending to have more NOx-saturated O3 production. This nonlinear effect also explains smaller O3 attribution to soil NO by the BFM than OSAT, especially over the NCP. Attributing a secondary pollutant to a primary emission (e.g., O3 to NO) is inherently tricky with nonlinear chemistry, as Koo et al. (2009) discussed. Therefore, it is useful to present estimates from different methods. The Path Integral Method (PIM) is a source apportionment method that explicitly treats nonlinear responses with mathematical rigor (Dunker et al., 2015). However, applying the PIM is more costly than the BFM or OSAT."

4. The conclusion of this paper is that the reduction of soil NO emission leads to the overall decrease of monthly MDA8 ozone concentration, and the reduction of ozone becomes more obvious with the increase of the reduction amount. NO can be titrated with O3. Does the author consider the depletion of ozone by NO?

Response: Yes. O3 titration by NO is considered in the model when we conducted different emission reduction scenarios. As illustrated by Fig. 6a, when soil NO emissions were reduced by 25%, slight increase (up to 1.3  $\mu$ g/m3) in the MDA8 ozone

concentration were observed over the North China Plain. These ozone increases reflect the effects of reduced  $O_3$  titration by NO for regions that are VOC-limited. However, as the reduction of soil NO emissions increases, the effect of reduced  $O_3$  titration is overwhelmed by reduced  $O_3$  formation due to less NO*x* available, thus leading to overall reduced ozone concentrations across China, as illustrated by Fig. 6b-d.

5. NOx and VOCs are important ozone precursors. Is it reasonable to consider only NO without considering the effect of VOCs on ozone concentration?

Response: We agree with the reviewer that both NO*x* and VOCs are important ozone precursors and both should be considered for ozone mitigation. However, the objective of the current study is to quantify the soil NO emissions and its impact on ozone concentrations in China, because soil NO emissions were generally considered as a "natural" source and was usually overlooked. In this study, we show that soil NO emissions represent an important contributor to ground-level ozone in China and reducing soil NO emissions could mitigate ozone pollution in China when NO*x* and VOCs emissions from other sources are unchanged. Our results also show that reducing soil NO emissions alone is not enough to eliminate ozone pollution completely, necessitating additional emission reductions from other sources, such as anthropogenic VOCs emissions. Follow-up studies should look at the synergistic effects of VOCs emissions (both anthropogenic and biogenic) and soil NO emissions on ozone concentration, as suggested by the reviewer.

6. In lines 195-196, the author pointed out that the spatial distribution of NO emissions in soil was very close to that of fertilization. However, I found that in the YRD area, there is a significant difference between the two, please explain the reason. And, Figure 4 of the article shows the simulation results of ozone concentration in China. In addition to the accidents in several hot cities introduced in the article, I noticed that the ozone concentration in the northwest of the Qinghai-Tibet Plateau was higher, please explain the reason. Refer to https://doi.org/10.1016/j.scitotenv.2022.160928.

Response: (1) The estimated soil NO emissions not only depend on the amount of applied fertilizer but also the meteorological conditions (e.g., temperature, precipitation) in the BDSNP algorithm. The figure below shows the spatial distribution of the soil NO emissions and fertilizer application zoomed over the YRD region (same as Fig. 2). The two spatial patterns generally match with a spatial correlation coefficient of 0.70.

**Figure R1.** Spatial distribution of (a) soil NO emissions for 2018 and (b) N and compound fertilizer applied for 2018 over the YRD region.

(2) According to the CAMx OSAT results (Fig. S8), the high ozone concentration over the Qinghai-Tibet Plateau is mostly contributed by the transport of boundary ozone, which includes both horizontal and vertical (i.e., stratosphere) directions. For regions with high altitude (e.g., the Qinghai-Tibet Plateau), vertical ozone intrusion from the stratosphere is most substantial, which is consistent with the finding by Chen et al. (2023) that the boundary layer height (BLH) was identified as the most important feature for ozone over the Qinghai-Tibet Plateau. Another study by Xue et al. (2011) indicated strong impact of anthropogenic forcing on the surface ozone on the Plateau due to air masses from the central and eastern China at Mount Waliguan in summer. However, our OSAT results suggest minimal ozone contribution from anthropogenic emissions over the Qinghai-Tibet Plateau (Fig. S8b). WRF simulated monthly mean wind directions over the northeast Qinghai-Tibet Plateau are mainly westerly (consistent with the back trajectories, Figure R2), again indicating negligible contribution from the central and eastern China. Relevant discussions have been added in the revised manuscript (L302-L309):

"Simulated ozone concentration over the northwest Qinghai-Tibet Plateau was also much higher than observed values. Our OSAT results (shown later) show that the high ozone concentration over the Qinghai-Tibet Plateau is mostly contributed by the transport of boundary ozone, which includes both horizontal and vertical (i.e., stratosphere) directions. For regions with high altitude (e.g., the Qinghai-Tibet Plateau), vertical ozone intrusion from the stratosphere is most substantial, which is consistent with the finding by Chen et al. (2023) that the boundary layer height was identified as the most important feature for ozone over the Qinghai-Tibet Plateau."

---

## Author Response (AR1)

**Response to reviewers' comments**

Reviewer #1:

In this paper, the authors used two methods to quantify the effect of soil NO emissions on surface ozone concentrations during the simulation period. The first is the traditional approach (BFM), which involves comparing simulated ozone concentrations in the base case with scenarios without soil NO emissions. The difference between the two scenarios is thought to represent the contribution of soil NO emissions to ozone. The second method uses the widely used ozone source allocation technique (OSAT) implemented by CAMx to label soil NO emissions as separate emission groups. The two methods basically reached the same conclusion: soil NO emission has a great effect on ozone concentration. This study emphasizes the importance of considering soil NO emissions in future ozone mitigation strategies in China. The article is integrative and innovative, and can be published. But it also has the following problems that need to be addressed by the authors.

Response: The authors thank the reviewer for reading the manuscript carefully and providing valuable comments. Our point-to-point response is given below. Revisions made to the manuscript and Supporting Information are highlighted in yellow.

**Major question:**

1. The Manuscript needs to add a discussion section in which the inclusion of comparisons with other models and the provision of comparative results with other relevant studies or models can provide a broader context for the application of the findings. More detailed information on the evaluation results and potential uncertainties in the model simulations would improve the robustness of the analysis.

Response: Thanks for the comment. We added a new section "3.4 Comparisons with existing studies" in the revised manuscript to specifically compare our estimated soil NO emissions as well as simulated results with existing studies (L450-L473):

*"3.4 Comparison with existing studies*

[revised manuscript text omitted]

2. For the simplicity of the article, some methods can be described briefly, but the main methods involved in the article need to be described carefully to enhance the readability of the article. The paper lacks the detailed introduction of WRF model configuration and model parameterization scheme. In addition, did the author consider the influence of regional differences in precipitation between north and south China on the model when selecting the parameterization scheme?

Response: Thank you for your suggestion. We listed the detailed model configurations in Table S3. We did not consider the influences of regional differences in precipitation when selecting the parameterization scheme since WRF does not permit different parameterization schemes within a single simulation.

**Table S3**. Model configurations of WRF and CAMx.

| Items | WRF | CAMx |
|---|---|---|
| Version | 3.7 | 7.0 |
| Gas-Phase chemistry | | Carbon Bond chemistry (CB06) (Yarwood et al., 2010) |
| Aerosol module | | CF Scheme / ISORROPIA |
| Aqueous-phase chemistry | | Updated mechanism of the Regional Acid Deposition Model (RADM) (Chang et al., 1987) |
| Planet boundary scheme | The Yonsei University Scheme (YSU) (Hong et al., 2006) | |
| Land surface | NOAH (Ek et al., 2003) | |
| Microphysics | Morrison double-moment scheme (Morrison et al., 2009) | |
| Shortwave radiation | RRTMG shortwave (Iacono et al., 2008) | |
| Longwave radiation | RRTMG scheme (Iacono et al., 2008) | |
| Boundary conditions / initial conditions | 1°×1° grids FNL Operational Global Analysis data archived at the GDAS | MOZART (Emmons et al., 2010) |
| Dry deposition and wet deposition | | Zhang et al. (2003) |

3. Please explain why the NO value simulated by OSAT model is higher than BFM

Response: We understand the reviewer is asking about the contribution of soil NO to $O_3$, as shown in Fig. 5. Differences in $O_3$ contribution estimates from OSAT and BFM arise from photochemical nonlinearity in the relationship between $O_3$ and NO emissions, as seen, for example, in Fig. 7a. This nonlinearity becomes stronger in regions with larger $NOx$ concentrations, especially where $O_3$ production is

characterized as NOx-saturated (or VOC-limited), such as the NCP. In such cases, removing a portion of the NO emissions (e.g., zeroing out soil NO for the BFM simulation) makes $O_3$ production from the remaining NO emissions more efficient, which lessens the $O_3$ response. Note that in Fig. 7a, the $O_3$ response for NCP is more curved (nonlinear) than other regions, consistent with NCP tending to have more NOx-saturated $O_3$ production. This nonlinear effect also explains smaller $O_3$ attribution to soil NO by the BFM than OSAT, especially over the NCP, as seen in Fig. 5. Attributing a secondary pollutant to a primary emission (e.g., $O_3$ to NO) is inherently tricky with nonlinear chemistry, as Koo et al. (2009) discussed, so it is useful to present estimates from different methods. The Path Integral Method (PIM) is a source apportionment method that explicitly treats nonlinear responses with mathematical rigor (Dunker et al., 2015). However, applying the PIM is more costly than the BFM or OSAT. We have added some discussions in the revised manuscript (L332-L346):

*"The difference between the two methods reflects the nonlinear ozone response to NOx emissions. This nonlinearity becomes stronger in regions with larger NOx concentrations, especially where $O_3$ production is characterized as NOx-saturated (or VOC-limited), such as the NCP. In such cases, removing a portion of the NO emissions (e.g., zeroing out soil NO for the BFM simulation) makes $O_3$ production from the remaining NO emissions more efficient, which lessens the $O_3$ response. As shown later in Fig. 7a, the $O_3$ response for NCP is more curved (nonlinear) than other regions, consistent with NCP tending to have more NOx-saturated $O_3$ production. This nonlinear effect also explains smaller $O_3$ attribution to soil NO by the BFM than OSAT, especially over the NCP. Attributing a secondary pollutant to a primary emission (e.g., $O_3$ to NO) is inherently tricky with nonlinear chemistry, as Koo et al. (2009) discussed. Therefore, it is useful to present estimates from different methods. The Path Integral Method (PIM) is a source apportionment method that explicitly treats nonlinear responses with mathematical rigor (Dunker et al., 2015). However, applying the PIM is more costly than the BFM or OSAT."*

4. The conclusion of this paper is that the reduction of soil NO emission leads to the overall decrease of monthly MDA8 ozone concentration, and the reduction of ozone becomes more obvious with the increase of the reduction amount. NO can be titrated with $O_3$. Does the author consider the depletion of ozone by NO?

Response: Yes. $O_3$ titration by NO is considered in the model when we conducted different emission reduction scenarios. As illustrated by Fig. 6a, when soil NO emissions were reduced by 25%, slight increase (up to 1.3 μg/m$^3$) in the MDA8 ozone

concentration were observed over the North China Plain. These ozone increases reflect the effects of reduced $O_3$ titration by NO for regions that are VOC-limited. However, as the reduction of soil NO emissions increases, the effect of reduced $O_3$ titration is overwhelmed by reduced $O_3$ formation due to less NO$x$ available, thus leading to overall reduced ozone concentrations across China, as illustrated by Fig. 6b-d.

5. NO$x$ and VOCs are important ozone precursors. Is it reasonable to consider only NO without considering the effect of VOCs on ozone concentration?

Response: We agree with the reviewer that both NO$x$ and VOCs are important ozone precursors and both should be considered for ozone mitigation. However, the objective of the current study is to quantify the soil NO emissions and its impact on ozone concentrations in China, because soil NO emissions were generally considered as a "natural" source and was usually overlooked. In this study, we show that soil NO emissions represent an important contributor to ground-level ozone in China and reducing soil NO emissions could mitigate ozone pollution in China when NO$x$ and VOCs emissions from other sources are unchanged. Our results also show that reducing soil NO emissions alone is not enough to eliminate ozone pollution completely, necessitating additional emission reductions from other sources, such as anthropogenic VOCs emissions. Follow-up studies should look at the synergistic effects of VOCs emissions (both anthropogenic and biogenic) and soil NO emissions on ozone concentration, as suggested by the reviewer.

6. In lines 195-196, the author pointed out that the spatial distribution of NO emissions in soil was very close to that of fertilization. However, I found that in the YRD area, there is a significant difference between the two, please explain the reason. And, Figure 4 of the article shows the simulation results of ozone concentration in China. In addition to the accidents in several hot cities introduced in the article, I noticed that the ozone concentration in the northwest of the Qinghai-Tibet Plateau was higher, please explain the reason. Refer to https://doi.org/10.1016/j.scitotenv.2022.160928.

Response: (1) The estimated soil NO emissions not only depend on the amount of applied fertilizer but also the meteorological conditions (e.g., temperature, precipitation) in the BDSNP algorithm. The figure below shows the spatial distribution of the soil NO emissions and fertilizer application zoomed over the YRD region (same as Fig. 2). The two spatial patterns generally match with a spatial correlation coefficient of 0.70.

[Figure]

**Figure R1.** Spatial distribution of (a) soil NO emissions for 2018 and (b) N and compound fertilizer applied for 2018 over the YRD region.

(2) According to the CAMx OSAT results (Fig. S8), the high ozone concentration over the Qinghai-Tibet Plateau is mostly contributed by the transport of boundary ozone, which includes both horizontal and vertical (i.e., stratosphere) directions. For regions with high altitude (e.g., the Qinghai-Tibet Plateau), vertical ozone intrusion from the stratosphere is most substantial, which is consistent with the finding by Chen et al. (2023) that the boundary layer height (BLH) was identified as the most important feature for ozone over the Qinghai-Tibet Plateau. Another study by Xue et al. (2011) indicated strong impact of anthropogenic forcing on the surface ozone on the Plateau due to air masses from the central and eastern China at Mount Waliguan in summer. However, our OSAT results suggest minimal ozone contribution from anthropogenic emissions over the Qinghai-Tibet Plateau (Fig. S8b). WRF simulated monthly mean wind directions over the northeast Qinghai-Tibet Plateau are mainly westerly (consistent with the back trajectories, Figure R2), again indicating negligible contribution from the central and eastern China. Relevant discussions have been added in the revised manuscript (L302-L309):

"*Simulated ozone concentration over the northwest Qinghai-Tibet Plateau was also much higher than observed values. Our OSAT results (shown later) show that the high ozone concentration over the Qinghai-Tibet Plateau is mostly contributed by the transport of boundary ozone, which includes both horizontal and vertical (i.e., stratosphere) directions. For regions with high altitude (e.g., the Qinghai-Tibet Plateau), vertical ozone intrusion from the stratosphere is most substantial, which is consistent with the finding by Chen et al. (2023) that the boundary layer height was identified as the most important feature for ozone over the Qinghai-Tibet Plateau.*"

[Figure]

**Figure S8.** Spatial distribution of ozone contribution from different source groups based on OSAT results.

[Figure]

**Figure R2.** WRF simulated monthly averaged wind speed and wind direction in June 2018 (left) and clusters of 72-hr backward trajectories at a location located in the Qinghai-Tiber Plateau (right). The backward trajectory clusters were generated by the TrajStat model.

**Minor questions:**

1. The caption of Figure 4 is not very clear. How are the results of observation and simulation represented in the figure? Does the author mean that scatter points indicate observations and base colors indicate simulation results?

Response: Yes. The scatter points represent observations over 365 cities in China while the base colors indicate the simulated ozone concentrations. The title of Figure 4 is modified in the revised manuscript for clarification. Revised title of Fig. 4:

*"Comparison of simulated (base colors) and observed (scatter points) values of MDA8 ozone in China in June 2018."*

2. (p11 Fig 5. Fig 6.) The map of soil NO ozone contribution from the Brute force method, and the model post-processing map can be done with map mask whitening.

Response: Thanks for pointing this out. We modified Fig. 5 and Fig. 6 with map mask whitening as suggested by the reviewer.

3. Some references have the problem of being too old, and newer research results can be added to the references.

Response: Thanks for pointing this out. We replaced some obsolete references with newer ones in the revised manuscript.

(L45-L49): Because high ozone concentration increases respiratory and circulatory risks (Malley et al., 2017; ***Cakaj et al., 2023; Wang et al., 2020***) and reduces crop yields (Feng et al., 2019; Lin et al., 2018; ***Mukherjee et al., 2021; Montes et al., 2022***), the coordinate control of $PM_{2.5}$ and $O_3$ was proposed as part of the 14th Five-year plan (State Council, 2021).

(L65-L68): However, NO$x$ emissions from soils (mainly as NO), as a result of microbial processes (e.g., nitrification and denitrification), could make up a substantial fraction of the total NO$x$ emissions (Lu et al., 2021; ***Drury et al., 2021***), yet is often overlooked.

(L76-L78) Soil NO emissions are affected by many factors, including nitrogen fertilizer application, soil organic carbon content, soil temperature, humidity, and pH (Vinken et al., 2014; Yan et al., 2005; ***Wang et al., 2021; Skiba et al., 2021***).

4. The resolution and clarity of all figures in the supplementary materials need to be improved. And, the results of ozone simulation in many areas of China exceed the boundary of China. Please confirm again whether it is reasonable.

Response: We have improved the resolution of the figures in the revised supplementary material. For the boundary issue, all the simulated ozone concentrations are in 36 km × 36 km grids. When we created the spatial plots, we masked out grid cells that do not cover any parts of China. For grid cells along the boundary of China that contain parts of China, we still kept them for plotting. So

visually the simulated ozone exceeds the boundary of China. This is simply due to the way of masking.

Reviewer #2:

General Comments:

This study adopted and applied the Berkeley-Dalhousie Soil NO$x$ Parameterization (BDSNP) algorithm to develop soil NO$x$ emission inventory, and evaluated their contribution to surface ozone concentration in China. Both the Brute-Force Method and Ozone Source Apportionment Technology were used in CAMx to assess spatial and temporal (diurnal and monthly) variations in five major regions of China. It is concluded that soil NO$x$ emissions substantially contributed to maximum daily 8-hr (MDA8) ozone concentrations by 8 to 12.5 ug/m$^3$ on average for June 2018, and led to an increase of exceedance days by 10 to 43.5% in selected regions. This study highlights the importance of soil NO$x$ in ozone formation in selected regions of China and suggests that control strategies aimed at soil NO$x$ emission reductions, along with other sectors, should be considered. This manuscript is well written, and the methodology used in, and the results derived from this study are scientifically sound but some improvement or clarification should be considered as follows:

Response: The authors thank the reviewer for reading the manuscript and providing helpful comments. We have revised the manuscript as suggested by the reviewer. Our point-to-point response is given below. Revisions made to the manuscript and Supporting Information are highlighted in yellow.

Specific Comments:

1.  Lines 37-39 (Abstract): It states that even with complete reductions in soil NO emissions about 450 million people are still exposed to unhealthy ozone levels, necessitating additional control policies such as transportation, power plant, etc. for selected regions. It is not clear whether or not the study suggests that soil NO$x$ emission reductions (e.g., use of nitrification inhibitors) are considered a priority over other NO$x$ emission sources.

Response: We do not conclude that soil NO$x$ emission reductions should be considered a priority over the other NO$x$ emission sources in this study. Instead, we showed that reducing soil NO emissions, which is not included in existing NO$x$ control policies, could be effective in reducing ozone concentrations and should also be considered in future mitigation strategies. The sentence that "*even with complete reductions in soil NO emissions, approximately 450.3 million people are still exposed to unhealthy ozone levels, necessitating additional control policies*" emphasizes the

equivalent importance of emission reductions from soil as well as other NO*x* emission sources. We revised the sentence as below to avoid any confusion (L494-L496):

*"However, even with the complete removal of soil NO emissions, approximately 450.3 million populations are still exposed to unhealthy ozone levels, necessitating multiple control policies at the same time."*

2.   Lines # 43-46 (Introduction): It states that with the substantial decrease in ambient fine PM concentrations, ozone emerges as a simultaneously targeted air pollutant. Please note that nitrogen oxides serve as important precursors to both tropospheric ozone and fine PM with consequent adverse effects so NO*x* control strategies would lead to reductions of both ozone and fine PM.

Response: Thanks for pointing this out. Due to the nonlinear ozone chemistry, the reductions in NO*x* emissions do not always lead to immediate ozone decrease. To avoid any confusion, we rewrite these sentences as below (L42-L49):

*"A substantial decrease in the atmospheric fine particulate matter (PM$_{2.5}$) concentrations has been witnessed during the past decade in China (Zhai et al., 2019; Xiao et al., 2020; Maji, 2020) while the ground-level ozone (O$_3$) concentrations do not exhibit a steady downward trend (Lu et al., 2020; Lu et al., 2021; Wang et al., 2022a; Sun et al., 2021). Because high ozone concentration increases respiratory and circulatory risks (Malley et al., 2017; Cakaj et al., 2023; Wang et al., 2020) and reduces crop yields (Feng et al., 2019; Lin et al., 2018; Mukherjee et al., 2021; Montes et al., 2022), the coordinate control of PM$_{2.5}$ and O$_3$ was proposed as part of the 14$^{th}$ Five-year plan (Council, 2021)."*

3.   Lines # 66-69 (Introduction): a number of references related to soil NO*x* with upper values were summarized here but other studies (e.g., https://agupubs.onlinelibrary.wiley.com/doi/full/10.1029/2020jd033304) should be acknowledged as well for completeness. For Sha et al, 2021 study, please note that in this study, the cropland regions included both high rates of fertilizer application and regular irrigation with the largest soil NO*x* of about 9 times higher than that of default in July.

Response: Thanks for pointing this out. We have added more relevant studies in the revised manuscript (P2, L68-L73).

*"In California, soil NOx emissions in July accounted for 40% of the state's total NOx emissions (when using an updated estimation algorithm) and resulted in 23% of enhanced surface ozone concentration (Sha et al., 2021). However, a wide range of*

*annual soil NOx emissions from 8,685 tons (as NO₂, (Guo et al., 2020)) to 161,100 metric tons of NOx-N (Almaraz et al., 2018) were reported depending on different methods."*

4. Lines # 109-143 (2.1. Estimation of soil NO emissions in China):

BDSNP estimates soil NO$x$ emissions based on available soil nitrogen, soil temperature, and soil moisture. This parameterization classifies each grid cell as either arid or non-arid and applies one of two static relationships between soil NO$x$ and soil moisture depending on that classification. Previous studies have shown that this relationship can be more dynamic, with emissions exhibiting different regional relationships between biome-specific NO emission factors (such as soil moisture) and soil NO$x$ emissions. Hence, the uncertainty of A'biome values can be significant so some general discussion is warranted in terms of the limitation of BDSNP capability in producing more self-consistent emissions estimates regardless of the choice of data input.

Response: Thanks for pointing this out. We added some relevant discussions in the revised manuscript (L245-L254):

*"Although the current BDSNP algorithm is considered more sophisticated than the old YL95 algorithm, it still suffers certain limitations. For example, the current BDSNP parameterization employs a static classification of "arid" versus "non-arid" soils, upon which the relationship between soil NO emissions and soil moisture relies (Hudman et al., 2012). However, recent studies (Sha et al., 2021; Huber et al., 2023) have shown more dynamic representation of this classification is needed to capture the emission characteristics as observed by many chamber and atmospheric studies (e.g., Oikawa et al. (2015); Huang et al. (2022)). Huber et al. (2023) also showed that the emission estimated based on the static classification are very sensitive to the soil moisture and thus could not produce self-consistent results when using different soil moisture products."*

5. Lines 116-122: What are the A'biome values used in the study, especially for cropland? How were these values determined? Suggest providing units for all terms in Eq. 1 here to help readers understand the physical meaning of the equation and the relationship of the terms included.

Response: Thanks for the suggestion. All relevant units have been added in the revised manuscript. We also rewrite "Section 2.1 Estimation of soil NO emissions in China" with more details on the BDSNP algorithm (L114-L131).

The A'biome is calculated as:

$$A'_{biome} = A_{w,biome} + N_{avail} \times \bar{E}$$

where $A_{w,biome}$ is the wet biome-dependent emission factor, which is based on Steinkamp and Lawrence (2011) (Table 5), as shown below. For cropland, $A_{w,biome}$ is 0.57 ng N m$^{-2}$ s$^{-1}$.

**Table 1**. Wet biome-dependent emission factor (unit: ng N$^{-2}$ s$^{-1}$, Steinkamp and Lawrence, 2011).

| ID | MODIS landcover | Köppen main climate* | Emission factor (wet) |
|---|---|---|---|
| 0 | Water | _ | 0 |
| 1 | Permanent wetland | _ | 0 |
| 2 | Snow and ice | _ | 0 |
| 3 | Barren | D, E | 0 |
| 4 | Unclassified | _ | 0 |
| 5 | Barren | A, B, C | $0.06^{+0.02}_{-0.02}$ |
| 6 | Closed shrubland | _ | $0.09^{+0.31}_{-0.07}$ |
| 7 | Open shrubland | A, B, C | 0.09 |
| 8 | Open shrubland | D, E | $0.01^{+0.00}_{-0.00}$ |
| 9 | Grassland | D, E | 0.84 |
| 10 | Savannah | D, E | $0.84^{+1.42}_{-0.53}$ |
| 11 | Savannah | A, B, C | $0.24^{+1.71}_{-0.21}$ |
| 12 | Grassland | A, B, C | $0.42^{+2.01}_{-0.35}$ |
| 13 | Woody savannah | _ | $0.62^{+0.57}_{-0.30}$ |
| 14 | Mixed forest | _ | $0.03^{+0.23}_{-0.03}$ |
| 15 | Evergr. broadl. forest | C, D, E | 0.36 |
| 16 | Dec. broadl. forest | C, D, E | $0.36^{+1.12}_{-0.27}$ |
| 17 | Dec. needlel. forest | _ | 0.35 |
| 18 | Evergr. needlel. forest | _ | $1.66^{+7.49}_{-1.36}$ |
| 19 | Dec. broadl. forest | A, B | $0.08^{+0.14}_{-0.05}$ |
| 20 | Evergr. broadl. forest | A, B | $0.44^{+2.27}_{-0.37}$ |
| 21 | Cropland | _ | $0.57^{+2.56}_{-0.46}$ |
| 22 | Urban and build-up lands | _ | 0.57 |
| 23 | Cropland/nat. veg. mosaic | _ | 0.57 |

*A: equatorial, B: arid, C: warm temperate, D: snow, E: polar.

6.   Line # 138-139: Please explain how this range of ‑147% to 69% was derived and its potential impacts on emission estimates.

Response: Thanks for point this out. The range of -147% to 69% represents the relative difference between the default fertilizer data based on International Fertilizer

Industry Association (IFA) fertilizer-use dataset for the year 2000 (Potter et al., 2010) and the provincial-level fertilizer data (including both pure nitrogen fertilizer and NPK compound fertilizer) obtained from statistical yearbook for year 2018. At the national scale, the total amount of pure nitrogen fertilizer from statistical yearbook for year 2018 was 20.7 Tg, which is only 5.6% higher than the default dataset. However, if we include the amount of NPK compound fertilizer and assume one-third of the compound fertilizer is nitrogen, the total amount of nitrogen fertilizer applied, according to statistical yearbook, is 28.2 Tg N, 43.9% higher than the default value. At provincial level, this discrepancy of fertilizer application ranges from -147% (Qinhai) to 69% (Xinjiang). Note that the relative difference for Tibet is over 700% because of relatively low amount of fertilizer applied. At regional level (see Table S1 for region definitions), the relative differences range from 9.1% (Southwest China) to 46.4% (Northwest). We updated these values in the revised manuscript (L145-L146):

*"At the regional level, the amount of total fertilizer differs by as much as 9.1% to 46.4% from the default fertilizer (Table S2)."*

**Table S2**. Differences of fertilizer applied between the default dataset and the statistical yearbook.

| Region/province | Amount of nitrogen fertilizer from Potter's dataset (Tg N) | Statistical yearbook for year 2018 | | | Relative difference (%) |
| --- | --- | --- | --- | --- | --- |
| | | Amount of pure nitrogen fertilizer (Tg N) | Amount of NPK compound fertilizer (Tg N) | Total (pure nitrogen fertilizer + 1/3 NPK fertilizer) (Tg N) | |
| Gansu | 0.48 | 0.33 | 0.27 | 0.42 | -13.7 |
| Ningxia | 0.12 | 0.16 | 0.15 | 0.21 | 45.6 |
| Qinghai | 0.11 | 0.03 | 0.03 | 0.05 | -146.7 |
| Shanxi | 0.59 | 0.89 | 0.99 | 1.22 | 51.2 |
| Xinjiang | 0.40 | 1.10 | 0.59 | 1.30 | 68.8 |
| *Northwest China* | *1.71* | *2.52* | *2.03* | *3.19* | *46.6* |
| Henan | 1.58 | 2.02 | 3.37 | 3.14 | 49.7 |
| Hubei | 0.91 | 1.13 | 1.08 | 1.49 | 38.6 |
| Hunan | 0.95 | 0.94 | 0.81 | 1.21 | 21.4 |
| *Central China* | *3.45* | *4.09* | *5.26* | *5.84* | *41.0* |

| | | | | | |
|---|---|---|---|---|---|
| Guizhou | 0.63 | 0.40 | 0.30 | 0.50 | -25.5 |
| Sichuan | 1.28 | 1.12 | 0.60 | 1.32 | 3.2 |
| Tibet | 0.19 | 0.01 | 0.02 | 0.02 | -728.3 |
| Yunnan | 0.74 | 1.05 | 0.56 | 1.24 | 40.5 |
| Chongqing | 0.46 | 0.46 | 0.25 | 0.54 | 14.8 |
| *Southwest China* | *3.30* | *3.05* | *1.74* | *3.63* | *9.1* |
| Guangdong | 0.75 | 0.89 | 0.71 | 1.12 | 33.3 |
| Guangxi | 0.84 | 0.74 | 0.95 | 1.06 | 20.4 |
| Hainan | 0.13 | 0.15 | 0.22 | 0.22 | 41.7 |
| *South China* | *1.72* | *1.77* | *1.88* | *2.40* | *28.4* |
| Anhui | 1.03 | 0.96 | 1.60 | 1.49 | 30.7 |
| Fujian | 0.34 | 0.42 | 0.31 | 0.52 | 35.5 |
| Jiangsu | 0.97 | 1.46 | 0.96 | 1.77 | 45.5 |
| Jiangxi | 0.67 | 0.34 | 0.53 | 0.52 | -29.9 |
| Shandong | 1.45 | 1.31 | 2.12 | 2.01 | 27.8 |
| Shanghai | 0.08 | 0.04 | 0.04 | 0.05 | -64.2 |
| Zhejiang | 0.48 | 0.40 | 0.23 | 0.48 | 0.0 |
| *East China* | *5.02* | *4.92* | *5.79* | *6.85* | *26.7* |
| Beijing | 0.09 | 0.03 | 0.03 | 0.04 | -106.6 |
| Hebei | 1.15 | 1.14 | 1.50 | 1.64 | 29.8 |
| Neimenggu | 0.74 | 0.86 | 0.77 | 1.12 | 33.6 |
| Shanxi | 0.48 | 0.25 | 0.64 | 0.47 | -2.6 |
| Tianjin | 0.05 | 0.06 | 0.08 | 0.08 | 40.7 |
| *North China* | *2.51* | *2.35* | *3.03* | *3.35* | *25.2* |
| Heilongjiang | 0.86 | 0.84 | 0.78 | 1.10 | 21.3 |
| Jilin | 0.51 | 0.58 | 1.50 | 1.08 | 52.8 |
| Liaoning | 0.53 | 0.55 | 0.68 | 0.78 | 31.9 |
| *Northeast China* | *1.90* | *1.97* | *2.96* | *2.95* | *35.6* |
| Total (China) | 19.60 | 20.65 | 22.69 | 28.22 | 30.5 |

Our emission calculation shows that if the default nitrogen dataset (i.e. Potter's value) is used, the estimated national total soil NO emissions are 805.2 Gg N/a for 2018, which is comparable to the value (770 Gg N/a averaged during 2008-2017) reported by Lu et al. (2021), but 30.5% lower than that based on the values obtained from the

statistical yearbook, which include both pure nitrogen fertilizer and compound fertilizer. Regionally, this underestimation ranges from 11.1%~41.5%, with a larger underestimation in Central China and East China (Figure. S3). These discussions were presented in L266-L271:

*"Our calculation shows that if only nitrogen fertilizer is considered, the estimated total soil NO emissions are 805.2 Gg N/a for 2018, which is comparable to the value (770 Gg N/a averaged during 2008-2017) reported by Lu et al. (2021), but 30.5% lower than that based on both nitrogen fertilizer and compound fertilizer. Regionally, this underestimation ranges from 11.1%~41.5%, with a larger underestimation in Central China and East China (Fig. S3)."*

[Figure]

**Figure S3.** Soil NO emissions estimated using different fertilizer data (default fertilizer data: International Fertilizer Industry Association from Potter et al. (2010) for year 2010; nitrogen fertilizer and compound fertilizer are from statistical yearbooks at the provincial level for year 2018).

6. Lines # 172-183 (2.3. Brute-force and OSAT): It is stated in the Introduction section that brute-force method might be inappropriate given the strong nonlinearity of the ozone chemistry but nevertheless it was used to simulate ozone concentration between the base case and a scenario case without soil NO emissions. Some discussion is warranted.

Response: Thanks for pointing this out. We have added relevant discussions in the revised manuscript (L332-L346).

*"The difference between the two methods reflects the nonlinear ozone response to NOx emissions. This nonlinearity becomes stronger in regions with larger NOx concentrations, especially where $O_3$ production is characterized as NOx-saturated (or VOC-limited), such as the NCP. In such cases, removing a portion of the NO*

*emissions (e.g., zeroing out soil NO for the BFM simulation) makes O₃ production from the remaining NO emissions more efficient, which lessens the O₃ response. As shown later in Figure 7a, the O₃ response for NCP is more curved (nonlinear) than other regions, consistent with NCP tending to have more NOx-saturated O₃ production. This nonlinear effect also explains smaller O₃ attribution to soil NO by the BFM than OSAT, especially over the NCP. Attributing a secondary pollutant to a primary emission (e.g., O₃ to NO) is inherently tricky with nonlinear chemistry, as Koo et al. (2009) discussed. Therefore, it is useful to present estimates from different methods. The Path Integral Method (PIM) is a source apportionment method that explicitly treats nonlinear responses with mathematical rigor (Dunker et al., 2015). However, applying the PIM is more costly than the BFM or OSAT."*

8. Lines # 200-201 (Fig. 2): I assume the authors used the same values of A'biome from Hudman et al. (2012). In this case, it would be 0.62% of available N in soil or 1.5% of fertilizer N applied for cropland. However, Fig. 2 indicated approximately 10% of fertilizer N was emitted as soil NO$x$ (judged on scales of the color schemes). Please explain the discrepancy between the emission rate (1.5% of the fertilizer N applied) of the original paper (Hudman et al., 2012) and that of this study (~10%). Additionally, NO$x$ is only one of many gaseous N species (NH3, N2O, NO$x$, and N2) emitted from fertilizer N from soils. Significant leaching/runoff losses (about 30% per IPCC defaults) exist. What are the expected N balances given the fact there are other more prominent N gas emissions with higher emission rates (NH3, N2O, and N2), leaching/runoff losses, and plant uptakes?

Response: Thanks for pointing this out. In Hudman et al. (2012), they reported a global soil NO emissions of 1.8 Tg N/yr due to fertilizer/manure N input (a total of 117 Tg N/yr), which is equivalent of 1.5% of applied fertilizer N, as pointed out by the reviewer. However, the total above-canopy soil NO emissions reported by Hudman et al. (2012) is 9.0 Tg N/yr, which is 7.7% of the applied fertilizer N. Figure 2a in our study shows the spatial distribution of the total above canopy soil NO emissions, which is estimated to be 1157.9 Gg N and accounts for 4.1% of the total amount of fertilizer N (28.2 Tg N) employed in this study. This ratio (4.1%) is slightly higher than that (3.9%) calculated from Lu et al. (2021), who reported 770 Gg N of soil NO emissions with 19.6 Tg N fertilizer input. It should be noted that this 4.1% ratio is a national average value. The ratio of soil NO emissions to applied fertilizer exhibits strong spatial heterogeneities (for example, more than 10% over croplands with higher rate of fertilizer application) that are dependent on the specific biome and the meteorological conditions.

As for the N balances, according to Zhang et al. (2021), reactive N ($N_r$) loss in China is defined as the sum of gas emissions, in the form of $NH_3$, NO$x$, and $N_2O$, plus leaching and runoff. $NH_3$, NO$x$, and $N_2O$ emissions were estimated to be 11.5, 4.5, and 1.8Tg N/year in 2018. N leaching and runoff accounts for 5.3 and 10.2 Tg N/year, respectively. These values sum up to a total of 33.3 Tg N/year Nr loss, among which N leaching/runoff accounts for 47% and NO$x$ emissions accounts for 13.5%. NO$x$ emissions from cropland and grassland in Zhang's study were only 0.6 Tg N/year (~50% lower than this study) and accounts for 1.8% of Nr loss.

Minor comments:

Pls use "Gg N/a" or "Gg N a-1" throughout the manuscript for consistency

Response: We have unified the unit as Gg N/a in the revised manuscript.

---

## Author Response (AR2)

Throughout the paper: The "x" in "NOx" is not a subscript. It should be. Please fix it.

Response: Modified in the revised manuscript.

L59-62: The myriad interactions between $NO_x$ and ozone should be discussed upfront in the introduction, e.g., how under high-$NO_x$ environments a reduction in $NO_x$ may enhance ozone via reduced ozone titration by NO and reduced OH titration by $NO_2$ (indeed Reviewer #1 also asked about it). Such nonlinearity of $NO_x$-ozone relationships are discussed elsewhere in the paper, but should be brought upfront in the introduction to set the stage for the rest of the paper.

Response: Thanks for pointing this out. We added relevant discussions in the introduction part (L55-L63):

*"The non-linear response of ozone formation to its precursors is well established (Kleinman et al., 1994; Sillman et al., 1990). In regions classified as $NO_x$-limited, reducing $NO_x$ emissions is an effective strategy for ozone mitigation. However, in regions classified as VOC-limited, typically characterized by high $NO_x$ emissions such as metropolitan areas, decreasing $NO_x$ emissions may actually result in increased ozone concentrations due to reduced ozone titration by NO and suppression of OH by $NO_2$ (Seinfeld and Pandis, 2016). Under such circumstances, reducing VOC emissions will counteract ozone increases caused by reducing $NO_x$ emissions."*

Sect. 3.1.2: It is unnecessary to capitalize "Uncertainties" in the heading.

Response: Corrected in the revised manuscript.

L260: If the ratio is 15:15:15, isn't it just 1:1:1? Does the number 15 carry any significant physical meaning? They do not add up to 100 either, so this is not clear why such a peculiar numerical ratio is used.

Response: In NPK compound fertilizer, the numbers such "15:15:15" represent the ratio of the three primary nutrients present in the fertilizer: nitrogen (N), phosphorus (P), and potassium (K). Each number represents the percentage of the nutrient by weight in the fertilizer. In the case of 15:15:15 NPK fertilizer, it means that the fertilizer contains 15% nitrogen, 15% phosphorus, and 15% potassium. The remaining percentage typically consists of other secondary nutrients, micronutrients, and filler materials. We have retained this terminology in the revised manuscript, as it is a customary convention for nomenclature. To avoid confusion, we have added explanations in the revised manuscript (L268-L270).

Sect. 3.3 and 3.4: As Reviewer #1 pointed out, in some regions soil NOx reductions can cause ozone to increase due to reduced oxidant titration. The authors responded correctly in the response file, but did not extend the discussion and explanation in the revised manuscript.

Indeed, the authors are recommended to discuss in greater detail why NOx reductions do not cause ozone to increase in high-NOx regions, which is what most previous studies have found. The comparison with previous studies, e.g., Shen et al. (2023), should also involve more explanation why the authors' results differ significantly from what's suggested by previous studies, highlighting the similarities and differences.

Response: Thank you for the suggestion. We have added relevant discussions in the revised manuscript (L398-L406):

*"With a 25% reduction in soil NO emissions, there was a widespread small decrease in monthly average MDA8 ozone concentration ($\Delta$MDA8: -1.5$\pm$0.9 $\mu g/m^3$), except over NCP where ozone showed a slight increase (up to 1.3 $\mu g/m^3$) in Shandong and Henan province. These ozone increases reflect the nonlinearity of ozone chemistry and this nonlinearity becomes stronger in regions with large $NO_x$ concentrations, especially where $O_3$ production is characterized as VOC-limited (such as NCP). When soil NO emissions were cut by 50%, the effect of reduced $O_3$ titration is overwhelmed by reduced $O_3$ formation due to less $NO_x$ available, thus the $\Delta$MDA8 showed a ubiquitous decrease across entire China with an average $\Delta$MDA8 of -5.5 $\mu g/m^3$."*

We also added the similarities and differences between our results and those of others in the revised manuscript (L482-L496):

*"The findings of this study align with previous studies, emphasizing the important role of soil NO emissions in influencing surface ozone concentrations in China. Furthermore, spatial heterogeneities exist in terms of both the soil NO emissions and the responses of ozone to reductions in soil NO emissions. However, it should be noted that the spatial pattern of ozone response to reduced soil NO emissions in this study is different from Shen et al. (2023). For instance, with a 30% reduction in soil NO emissions, $O_3$ concentration increased by 3-5 ppb over Inner Mongolia, Heilongjiang, Xinjiang, and Tibet and decreased by 0-2 ppb over the Yangtze River basin in Shen et al. (2023). In this study, a 20% reduction in soil NO emissions was found to lead to widespread but small decrease (less than 4 $\mu g/m^3$) in ozone concentrations except the NCP (Fig. 6a). These inconsistences may stem from the differences in the estimated soil NO emissions, both associated with the magnitude and the spatial distribution, as also noted in other study (Zhu et al., 2023). Therefore, more observations, such as direct measurement of soil NO flux, especially over agricultural areas, are urgently needed to better constrain the estimated soil NO emissions."*

Concluding section
Every article must have a final section where the overall advances are concisely summarised and put in context. Although the results section may include some discussion, a synthesis and

interpretation must appear in the final section. ACP expects that the concluding section will normally include the following components, although not necessarily in separate paragraphs:

Summary: Summarise the main results and relate them to the objectives, questions or hypotheses of the study. The summary should include the main quantitative results.

Synthesis/interpretation: Explain and interpret the results concisely to enable readers to make sense of them as a whole.

Comparison and context: Compare the results with previous studies to put them in context. Explain consistencies, inconsistencies and advances in knowledge.

Caveats and limitations: State how these affect confidence in the overall results, and where future work is needed.

Implications: Discuss what the results mean for our understanding of the state and/or behaviour of the atmosphere and climate, which is the main requirement for publication in ACP. The editor's acceptance/rejection decision will be strongly guided by this component of the concluding section.

Response: Thanks for the guidance. Our revised conclusion part is as following (L497-L526):

"Soil NO emissions are non-negligible $NO_x$ sources, particularly during summer. The importance of soil NO emissions to ground-level ozone in China is much less evaluated than combustion $NO_x$ emissions. In this study, the total national soil NO emissions were estimated to be 1157.9 Gg N in 2018 based the BDSNP algorithm, with a spatial distribution closely following that of fertilizer application. High soil NO emissions were greatest over Henan, Shandong, and Hebei provinces, which differs significantly from where anthropogenic $NO_x$ emissions are. Distinct diurnal and seasonal variations in soil NO emissions were found, mainly driven by the changes in soil temperature as well as the timing of fertilizer application. Uncertainty analysis of the estimated soil NO emissions reveals a range of 715.7~1902.6 Gg N that warrants further study and, preferably, constraint from observations.

Using two ozone source attribution methods (BFM and OSAT), we evaluated the contribution of soil NO emissions to ground-level ozone concentration for June 2018. Both methods suggest a substantial contribution of soil NO emissions to MDA8 ozone concentrations of 8~12.5 $\mu g/m^3$ on average for June 2018, with the OSAT results consistently higher than BFM. Soil NO emissions were shown to increase of ozone exceedances days (i.e., MDA8 above 160 $\mu g/m^3$) by 10.0%~43.5% depending on region. Reducing soil NO emissions could generally reduce the ground-level ozone concentrations and population exposure to unhealthy ozone levels, especially over NCP and YRD. For example, a 50% reduction in soil NO emissions decreased land area experiencing ozone above 160 $\mu g/m^3$ by 15.2% and the population exposed to this ozone concentration by 8.0%. However, even with complete removal of soil NO emissions, approximately 450.3 million people are still exposed to ozone above 160 $\mu g/m^3$.

*The major findings of this study reinforce previous studies by highlighting the important contribution of soil NO emissions to surface ozone concentrations in China, although substantial uncertainties remain with soil NO emission estimates. Observational constraints on the magnitude of soil $NO_x$ emissions in China are needed. Ozone response to reducing soil NO emissions varies by region due to the non-linear chemistry of ozone formation. Future ozone mitigation strategies should consider the potential benefit of reducing non-combustion $NO_x$ emissions, such as soil NO, with due consideration to the sensitivity of ozone to reducing $NO_x$ in the region."*

**Reference:**

Kleinman, L., Lee, Y. N., Springston, S. R., Nunnermacker, L., Zhou, X., Brown, R., Hallock, K., Klotz, P., Leahy, D., and Lee, J. H.: Ozone formation at a rural site in the southeastern United States, Journal of Geophysical Research: Atmospheres, 99, 3469-3482, 1994.

Seinfeld, J. H. and Pandis, S. N.: Atmospheric chemistry and physics: from air pollution to climate change, John Wiley & Sons2016.

Shen, Y., Xiao, Z., Wang, Y., Xiao, W., Yao, L., and Zhou, C.: Impacts of agricultural soil NOx emissions on O3 over Mainland China, Journal of Geophysical Research: Atmospheres, e2022JD037986, 2023.

Sillman, S., Logan, J. A., and Wofsy, S. C.: The sensitivity of ozone to nitrogen oxides and hydrocarbons in regional ozone episodes, Journal of Geophysical Research: Atmospheres, 95, 1837-1851, 1990.

Zhu, Q., Place, B., Pfannerstill, E. Y., Tong, S., Zhang, H., Wang, J., Nussbaumer, C. M., Wooldridge, P., Schulze, B. C., and Arata, C.: Direct observations of NO x emissions over the San Joaquin Valley using airborne flux measurements during RECAP-CA 2021 field campaign, Atmospheric chemistry and physics, 23, 9669-9683, 2023.